# AI in Thyroid Cancer Diagnosis: Techniques, Trends, and Future Directions

Yassine Habchi [1], Yassine Himeur [2,*,†], Hamza Kheddar [3,†], Abdelkrim Boukabou [4,†], Shadi Atalla [2,†], Ammar Chouchane [5,†], Abdelmalik Ouamane [6,†] and Wathiq Mansoor [2,†]

[1] Institute of Technology, University Center Salhi Ahmed, BP 58 Naama, Naama 45000, Algeria; habchi@cuniv-naama.dz
[2] College of Engineering and Information Technology, University of Dubai, Dubai 14143, United Arab Emirates; satalla@ud.ac.ae (S.A.); wmansoor@ud.ac.ae (W.M.)
[3] LSEA Laboratory, Electrical Engineering Department, University of Medea, Medea 26000, Algeria; hamza.kheddar@gmail.com
[4] Department of Electronics, University of Jijel, BP 98 Ouled Aissa, Jijel 18000, Algeria; aboukabou@univ-jijel.dz
[5] Department of Electrical Engineering, University of Yahia Fares Medea, Medea 26000, Algeria; chouchane.ammar@univ-medea.dz
[6] Laboratory of LI3C, Mohamed Khider University, Biskra 07000, Algeria; ouamaneabdealmalik@univ-biskra.dz
* Correspondence: yhimeur@ud.ac.ae
† These authors contributed equally to this work.

**Abstract:** Artificial intelligence (AI) has significantly impacted thyroid cancer diagnosis in recent years, offering advanced tools and methodologies that promise to revolutionize patient outcomes. This review provides an exhaustive overview of the contemporary frameworks employed in the field, focusing on the objective of AI-driven analysis and dissecting methodologies across supervised, unsupervised, and ensemble learning. Specifically, we delve into techniques such as deep learning, artificial neural networks, traditional classification, and probabilistic models (PMs) under supervised learning. With its prowess in clustering and dimensionality reduction, unsupervised learning (USL) is explored alongside ensemble methods, including bagging and potent boosting algorithms. The thyroid cancer datasets (TCDs) are integral to our discussion, shedding light on vital features and elucidating feature selection and extraction techniques critical for AI-driven diagnostic systems. We lay out the standard assessment criteria across classification, regression, statistical, computer vision, and ranking metrics, punctuating the discourse with a real-world example of thyroid cancer detection using AI. Additionally, this study culminates in a critical analysis, elucidating current limitations and delineating the path forward by highlighting open challenges and prospective research avenues. Through this comprehensive exploration, we aim to offer readers a panoramic view of AI's transformative role in thyroid cancer diagnosis, underscoring its potential and pointing toward an optimistic future.

**Keywords:** thyroid carcinoma detection; thyroid cancer segmentation; machine learning; deep learning; convolutional neural networks

## 1. Introduction

### 1.1. Background

The adoption of AI in healthcare has become a pivotal development, profoundly reshaping the landscape of medical diagnosis, treatment, and patient care. AI's exceptional capabilities, including pattern recognition, predictive analytics, and decision-making skills, enable the development of systems that can analyze complex medical data at a scale and precision beyond human capacity [1]. This, in turn, augments early disease detection, facilitates accurate diagnoses, and aids personalized treatment planning. Moreover, AI-driven predictive models can forecast disease outbreaks, enhance the efficiency of hospital

operations, and significantly improve patient outcomes [2]. Additionally, AI has the potential to democratize healthcare by bridging the gap between rural and urban health services and making high-quality care more accessible. Hence, the importance of AI in healthcare is profound and will continue to grow as technology advances, leading to more sophisticated applications and better health outcomes for patients worldwide [3,4]. However, the trust serves as a mediator, influencing the impact of AI-specific factors on user acceptance. Researchers have investigated how security, risk, and trust impact the adoption of AI-powered assistance [5]. They have conducted empirical tests on their proposed research framework and found that trust plays a pivotal role in determining user acceptance.

Cancer, a leading cause of death, affects various body parts, as depicted in Figure 1a. Among different types, thyroid carcinoma is one of the most common endocrine cancers globally [6,7]. Concerns are mounting over the escalating incidence of thyroid cancer and associated mortality. Research indicates that thyroid cancer incidence is higher in women aged 15–49 (ranked fifth globally) than in men aged 50–69 years [8–10].

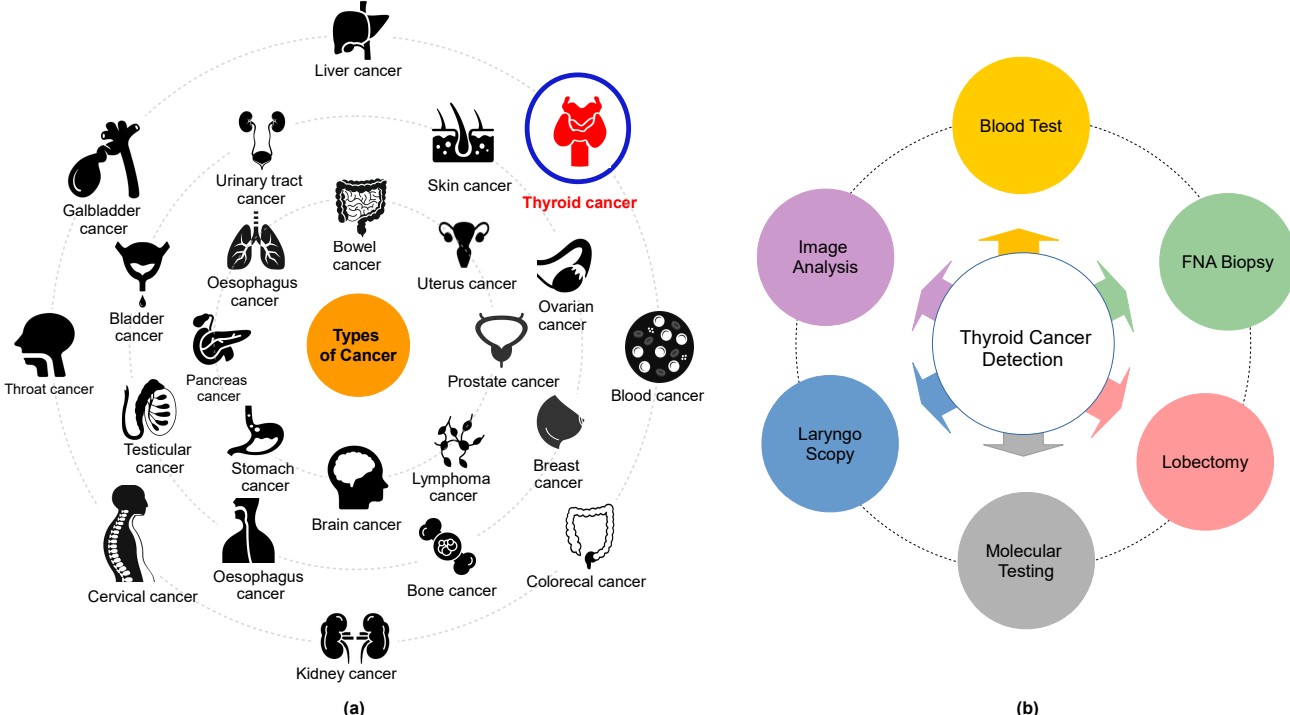

**Figure 1.** (**a**) Some of the common types of cancer and (**b**) thyroid cancer detection methods.

According to existing global epidemiological data, the rapid growth of abnormal thyroid nodules is driven by an accelerated increase in genetic cell activity, where the normal functioning and activity of cells in an organism are heightened or intensified. This condition can be categorized into four primary subtypes: papillary carcinoma (PTC) [11], follicular thyroid carcinoma (FTC) [12], anaplastic thyroid carcinoma (ATC) [13], and medullary thyroid carcinoma (MTC) [14]. Influential factors such as high radiation exposure, Hashimoto's thyroiditis, psychological and genetic predispositions, and advancements in detection technology can contribute to the onset of these cancer types. These conditions might subsequently lead to chronic health issues, including diabetes, irregular heart rhythms, and blood pressure fluctuations [15–17]. Although the quantity of cancer cells is a significant indicator for assessing both invasiveness and poor prognosis in thyroid carcinoma, obtaining results is often time-consuming due to the requirement to observe cell appearance. Therefore, the detection and quantification of cell nuclei are considered alternative biomarkers for assessing cancer cell proliferation.

The utilization of computer-aided diagnosis (CAD) systems for analyzing thyroid cancer images has seen a significant increase in popularity in recent years. These systems, renowned for enhancing diagnostic accuracy and reducing interpretation time, have become an invaluable tool in the field. Among these technologies, radionics, when used in conjunction with ultrasonography imaging, has become widely accepted as a cost-effective, safe, simple, and practical diagnostic method in clinical practice. Endocrinologists frequently conduct US scans in the 7–15 MHz range to identify thyroid cancer and evaluate its anatomical characteristics. The American College of Radiology has formulated a Thyroid Imaging, Reporting, and Data System (TI-RADS) that classifies thyroid nodules into six categories based on attributes such as composition, echogenicity, shape, size, margins, and echogenic foci. These classifications range from normal (Thyroid Imaging, Reporting, and Data System (TIRADS)-1) to malignant (TIRADS-6) [18–20]. Several open-source applications are available for assessing these thyroid cancer features [21,22]. However, the identification and differentiation of nodules continue to present a challenge, largely reliant on radiologists' personal experience and cognitive abilities. This is due to the subjective nature of human eye-based image recognition, the poor quality of captured images, and the similarities among US images of benign thyroid nodules, malignant thyroid nodules, and lymph nodes.

Moreover, ultrasonography imaging is often a time-intensive and stressful procedure, which can result in inaccurate diagnoses. Misclassifications among normal, benign, malignant, and indeterminate cases are typical [23–28]. A fine-needle aspiration biopsy (FNAB) is typically conducted for a more precise diagnosis. However, FNAB can be an uncomfortable experience for patients, and a specialist's lack of knowledge can potentially convert benign nodules into malignant ones, not to mention the additional financial burden [29,30] (refer to Figure 1b). Selecting their characteristics is the primary challenge in distinguishing between benign and malignant nodules. Numerous studies have explored the characterization of conventional US imaging for various types of cancers, including retina [31,32], breast cancer [33–37], blood cancer [38,39], and thyroid cancer [40,41]. However, these methods remain insufficiently accurate for the reliable classification of thyroid nodules.

The incorporation of AI technology plays a pivotal role in reducing subjectivity and enhancing the accuracy of pathological diagnoses for various intractable diseases, including those affecting the thyroid gland [42,43]. This enhancement is achieved through an improved interpretation of ultrasonography images and faster processing times. Machine learning (ML) and deep learning (DL) have surfaced as potential solutions for automating the classification of thyroid nodules in applications such as US, fine-needle aspiration (FNA), and thyroid surgery [44,45]. This potential has been underscored in numerous studies, such as [43,46–50]. Furthermore, there are ongoing studies examining the use of this innovative technology for cancer detection, where its effectiveness hinges on the volume of data and the precision of the classification process.

The motivation to write a review on "AI in thyroid cancer diagnosis" stems from the increasing prevalence of thyroid cancer, a significant endocrine malignancy where early and accurate detection is pivotal for patient outcomes. As technological advancements in AI and machine learning burgeon, their integration into medical diagnostics—spanning imaging, pathology, and genomics—offers potential improvements in detection accuracy and efficiency. Traditional thyroid carcinoma diagnostic methods, such as fine-needle aspiration biopsies, sometimes present inconclusive results; AI promises less-invasive alternatives with possibly superior precision. This review amalgamates insights from the intersection of computer science, radiology, pathology, and endocrinology, propelling multidisciplinary collaboration. It also spotlights AI's clinical implications, guiding clinicians in leveraging its capabilities for patient care, while delineating future research directions. Furthermore, this review underscores the economic and healthcare benefits, from cost savings to reduced waiting times. At the same time, it is imperative to address AI's inherent challenges, including data privacy and ethical considerations, ensuring its balanced integration into healthcare. In essence, the review offers a comprehensive panorama of AI's

current and potential role in thyroid carcinoma detection, benefiting both researchers and medical practitioners.

### 1.2. Our Contribution

This review provides a comprehensive examination of the application of AI methods in detecting thyroid cancer. The objective of AI-based analysis in the medical field is increasingly shifting towards enhancing diagnostic accuracy, and this review aims to illustrate this trend, particularly in thyroid cancer detection. We first provide an overview of the existing frameworks and delve into the specifics of various AI techniques. These include supervised learning methods, such as DL, artificial neural networks, traditional classification, and PMs, as well as USL methods, such as clustering and dimensionality reduction. We also explore ensemble methods, including bagging and boosting. Recognizing the importance of quality datasets in AI applications, we scrutinize several TCDs, addressing their features, as well as feature selection and extraction methods used in various studies. We then outline the standard assessment criteria used to evaluate the performance of AI-based thyroid cancer detection methods. These range from classification and regression metrics to statistical metrics, machine vision metrics, and ranking metrics. Finally, we discuss future research directions, emphasizing areas that require more attention to overcome existing barriers and improve the use and deployment of thyroid cancer detection solutions. In conclusion, we underscore the potential of AI in advancing thyroid cancer detection while also noting the need for a continuous critical evaluation to ensure its responsible and effective use.

Accordingly, the principal contributions of our paper are as follows:

- An overview of existing frameworks and specifics of various AI techniques, including supervised learning (DL, artificial neural networks, traditional classification, and PMs) and USL (clustering and dimensionality reduction) methods, as well as ensemble methods (bagging and boosting).
- An examination of multiple TCDs, exploring the characteristics of these datasets, as well as the methods employed for selecting and extracting features in different research studies.
- An outline of standard assessment criteria used to evaluate the performance of AI-based thyroid cancer detection methods, encompassing classification and regression metrics, statistical metrics, computer vision metrics, and ranking metrics.
- A critical analysis and discussion highlighting limitations, hurdles, current trends, and open challenges in the field.
- A discussion of future research directions, emphasizing areas requiring more attention to overcome existing barriers and improve thyroid cancer detection solutions.
- An emphasis on the potential of AI in advancing thyroid cancer detection while advocating a continuous critical evaluation for responsible and effective use.

Additionally, the principal contributions of the proposed review compared to other existing surveys are summarized in Table 1.

**Table 1.** The significant contributions of the proposed review on thyroid cancer classification in comparison with other related studies.

| Ref | Year | PPY | TCDS | AIA | Open Challenges | | | Future Directions | | | | | |
|-----|------|-----|------|-----|------|------|----|-----|--------|----|----|------|----|
| | | | | | TCDA | RDLA | PP | XAI | EFC-AI | RL | PS | IoMIT | RS |
| [51] | 2021 | ✓ | ✓ | ✗ | ✗ | ✗ | ✗ | ✗ | ✗ | ✗ | ✗ | ✗ | ✗ |
| [52] | 2021 | ✓ | ✓ | ✗ | ✗ | ✗ | ✗ | ✗ | ✗ | ✗ | ✗ | ✗ | ✗ |
| [53] | 2021 | ✓ | ✓ | ✗ | ✗ | ✗ | ✗ | ✗ | ✗ | ✗ | ✗ | ✗ | ✗ |
| [54] | 2021 | ✓ | ✓ | ✗ | ✗ | ✗ | ✗ | ✗ | ✗ | ✗ | ✗ | ✗ | ✗ |
| [55] | 2021 | ✓ | ✓ | ✗ | ✗ | ✗ | ✗ | ✗ | ✗ | ✗ | ✗ | ✗ | ✗ |
| [56] | 2022 | ✓ | ✓ | ✗ | ✗ | ✗ | ✗ | ✗ | ✗ | ✗ | ✗ | ✗ | ✗ |
| [57] | 2022 | ✓ | ✓ | ✗ | ✗ | ✗ | ✗ | ✗ | ✗ | ✗ | ✗ | ✗ | ✗ |
| [58] | 2022 | ✓ | ✓ | ✗ | ✗ | ✗ | ✗ | ✗ | ✗ | ✗ | ✗ | ✗ | ✗ |
| [59] | 2022 | ✓ | ✓ | ✗ | ✗ | ✗ | ✗ | ✗ | ✗ | ✗ | ✗ | ✗ | ✗ |
| Ours | - | ✓ | ✓ | ✓ | ✓ | ✓ | ✓ | ✓ | ✓ | ✓ | ✓ | ✓ | ✓ |

Abbreviations: edge, fog, and cloud networks based on AI (EFC-AI).

### 1.3. Road Map

The rest of this paper is organized as follows. Section 2 follows, providing an overview of existing frameworks utilized in this field, and discussing their respective advantages and limitations. Section 3 presents various TCDs used in AI-based analyses, explaining their relevance and uniqueness. In Section 4, the paper delves into the vital aspect of "Features", discussing feature extraction and selection methods in AI models used for thyroid cancer detection. Section 5 outlines the standard assessment criteria used to evaluate the performance of these models. An actual instance of AI-based thyroid cancer detection is presented in Section 6 to provide a real-world context to the theoretical aspects discussed earlier. The paper then proceeds to a critical analysis and discussion in Section 7, where challenges, limitations, and areas for improvement in the current approaches are discussed. In Section 8, potential future research directions are proposed, highlighting areas where further exploration and innovation can lead to advancements in AI-based thyroid cancer detection. The paper concludes with Section 9, summarizing the main findings and discussions, thereby providing a comprehensive conclusion to the discussions presented in the earlier sections.

## 2. Overview of Existing Frameworks

This section showcases the various AI-based methods utilized for diagnosing thyroid gland (TG) cancers. In the illustration, Figure 2 presents a proposed categorization of the thyroid cancer diagnosis techniques relying on AI.

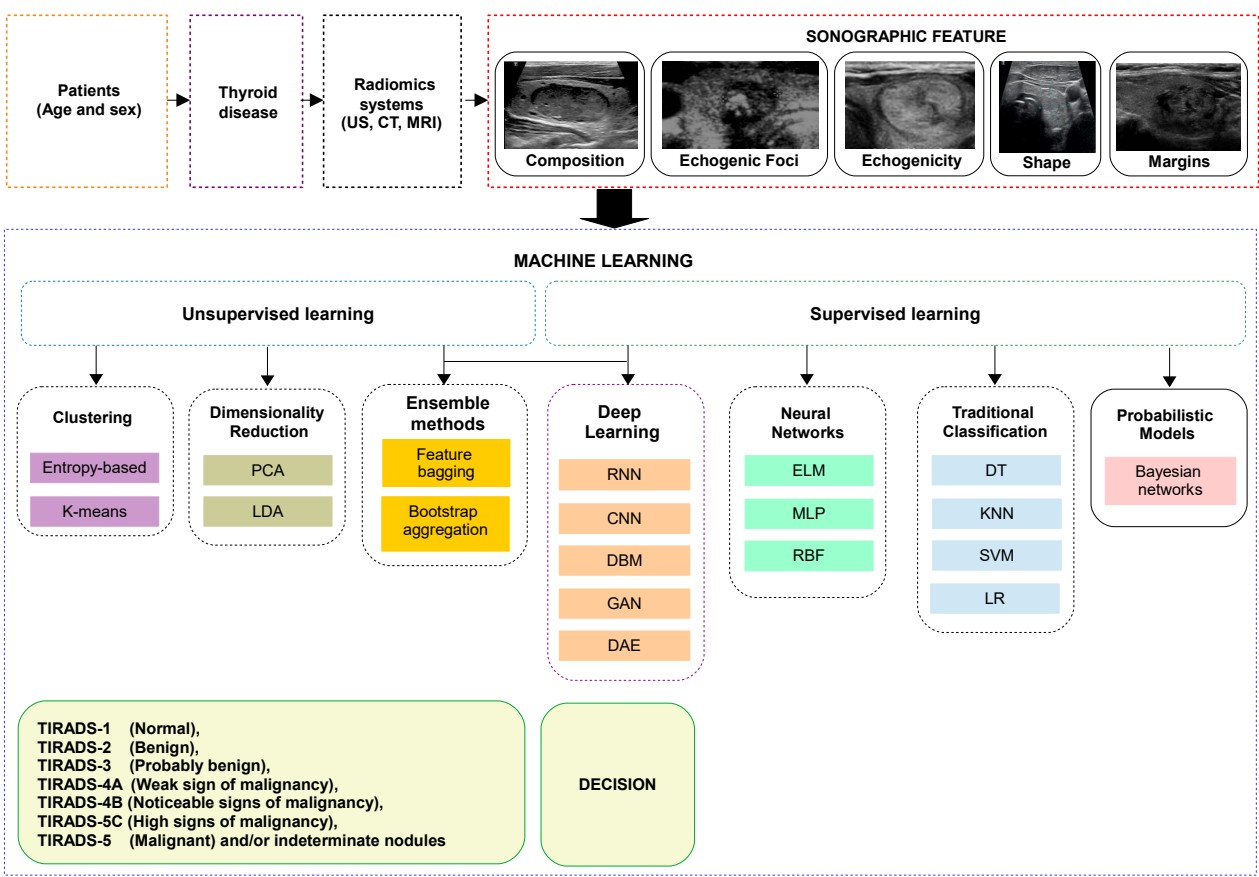

**Figure 2.** Taxonomy of the thyroid cancer detection schemes based in AI.

### 2.1. Objective of AI-Based Analysis (O)

This article focuses on the application of AI in thyroid cancer detection. In order to better understand the purpose behind each framework, it is crucial to identify the objective of each approach.

**O1. Classification**: Thyroid carcinoma classification refers to the categorization of thyroid cancers based on their histopathological features, clinical behavior, and prognosis. There are several types of thyroid carcinomas, each of which has distinct characteristics. The primary categories include: (i) PTC: The most common type, accounting for about 80% of all thyroid cancers. PTC tends to grow very slowly, but it often spreads to lymph nodes in the neck. Despite this, it is usually curable with treatment; (ii) FTC: the second most common type, FTC can invade blood vessels and spread to distant parts of the body, but it is less likely to spread to lymph nodes; (iii) MTC: This type of thyroid cancer starts in the thyroid's parafollicular cells, also called C cells, which produce the hormone calcitonin. Elevated levels of calcitonin in the blood can indicate MTC; (iv) ATC: A very aggressive and rare form of thyroid cancer, ATC often spreads quickly to other parts of the neck and body. It is difficult to treat.

The classification of thyroid carcinomas is crucial in determining the most effective course of treatment for each patient. Various factors such as tumor size, location, and the patient's age and overall health are also taken into consideration when forming a treatment plan. Advances in AI and machine learning are helping to automate and improve the accuracy of thyroid carcinoma classification, with many models trained to classify tumors based on medical images or genetic data. Liu et al. [60] report on incorporating a support vector machine (SVM) for cancer detection. Similarly, Zhang et al. [61,62] propose deep neural network (DNN)-based strategies for segregating and categorizing benign and malignant thyroid nodules in ultrasound imagery. Furthermore, the bidirectional LSTM (Bi-LSTM) model, as presented by Chen et al. [63], demonstrates a notable accuracy in classifying thyroid nodules. These classification systems constitute structured hierarchies instrumental in organizing knowledge and workflows in the specific domain of thyroid cancer.

**O2. Segmentation**: The segmentation of thyroid carcinoma refers to the process of identifying and delineating the region of an image that corresponds to a thyroid tumor. The goal of segmentation is to separate the areas of interest, in this case, the thyroid tumor, from the surrounding tissues in medical images. This can be done manually by an expert radiologist, or it can be automated using machine learning algorithms [64,65]. Segmentation is a crucial step in medical image analysis because it helps to accurately determine the location, size, and shape of the tumor, which are vital parameters for diagnosis, treatment planning, and prognosis prediction. A variety of methods can be used to perform image segmentation, including thresholding, edge detection, region-growing methods, and more complex machine learning and DL techniques.

In the case of thyroid carcinoma, the segmentation can be challenging due to the high variability in the appearance and shape of the tumors, their proximity to other structures in the neck, and the presence of noise or artifacts in the images. Therefore, robust and reliable segmentation algorithms are needed to ensure accurate and consistent results. AI methods, including convolutional neural networks (CNNs) and the U-Net architecture, are being increasingly used for thyroid carcinoma segmentation because of their ability to learn and generalize from large quantities of data, thus improving the accuracy and reliability of the segmentation process.

**O3. Prediction**: The prediction of thyroid carcinoma involves the use of various diagnostic tools, tests, and techniques—often employing machine learning models—to anticipate the probability of a patient developing thyroid cancer. This predictive analysis can be based on several factors, including but not limited to (i) genetic predisposition: individuals with a family history of thyroid cancer are at a higher risk; (ii) gender and age: thyroid cancer is more common in women and people aged between 25 and 65; (iii) radiation exposure: exposure to high levels of radiation, especially during childhood, increases the risk of developing thyroid cancer; (iv) diet and lifestyle: a lack of iodine in the diet and certain lifestyle factors may contribute to an increased risk. In a medical context, prediction does not necessarily mean a certain future outcome, but rather it points to an increased risk or likelihood based on current data and predictive models. For thyroid carcinoma, predictive tools and tests are typically used in conjunction with each other to achieve more accurate

results. For instance, machine learning algorithms can be trained on historical medical data to predict the likelihood of a nodule being benign or malignant, aiding in early detection and more effective treatment planning. Various studies have been proposed to predict thyroid cancer. For instance, in [66], the authors employed the use of an artificial neural network (ANN) and a logistic regression (LR) to make predictions. Another study [67] detailed the creation of a predictive machine using a CNN to analyze 10,068 microscopic thyroid cancer images from South Asian populations. The thyroid cancer images were a part of pharmacogenomic datasets, encompassing genomics and a variation analysis of individual differences associated with the predisposition to the disease.

### 2.2. Preprocessing

Dimensionality reduction (DR) is a technique used in the field of ML, particularly in the preprocessing and feature engineering phases, that transforms data from a high-dimensional space into a lower-dimensional space. This technique is popular for classification due to its cost-effectiveness and ability to eliminate unnecessary data patterns and minimize redundancy. For instance, DR was used to diagnose thyroid disease (TD) using cytological images [68].

Principal component analysis (PCA) is a multivariate statistical preprocessing method that transforms variables into a reduced set of uncorrelated variables. This approach reduces the number of variables and minimizes redundant information while preserving the relationships among the data as much as possible. PCA has been widely used in cancer detection and classification of benign and malignant thyroid cells. For example, in [69], PCA was utilized to select the optimal set of wavelet coefficients from the application of the double-tree complex wavelet transform (DTCW) on noisy thyroid images, which were then classified using a random forest (RF). In [70], PCA was applied to data from 399 patients with three types of thyroid carcinoma (papillary, follicular, and undifferentiated) in Morocco, enabling a classification based on factors such as sex, age, type of carcinoma, and region.

### 2.3. Supervised Learning (SL)

SL is a method of machine learning where an algorithm is trained to classify or predict the condition based on labeled data, which, in this case, are medical data related to thyroid cancer. The aim of supervised learning is to differentiate between the different forms of thyroid cancer through the use of annotated data and examples. For example, these data can include ultrasound images, radiomic features, genetic markers, patient demographics, or any other information that may be relevant to the diagnosis or prognosis of thyroid cancer. The labeled data may indicate whether each instance corresponds to a case of thyroid cancer or not, or it may provide more detailed labels such as the stage of the cancer or the type of thyroid carcinoma.

In a classification setting, the supervised learning algorithm could be trained to distinguish between benign and malignant thyroid nodules based on certain characteristics extracted from medical imaging data. The labels in the training data would specify whether each nodule is benign or malignant. After training, the algorithm can then be used to classify new, unlabeled nodules. Similarly, a regression-based supervised learning algorithm might be trained to predict the progression or the prognosis of thyroid cancer based on various patient-specific features. The labels here would correspond to a continuous outcome variable, such as the survival time of the patient or a measure of disease progression. It is important to note that the performance of these methods heavily relies on the quality and quantity of the available data. The more accurate and comprehensive the data, the better the algorithm will perform in predicting or classifying new instances. Additionally, supervised learning models in healthcare, including thyroid carcinoma detection, need to be validated on separate test datasets and in real-world clinical settings to ensure their robustness and reliability [71,72].

Traditional Classification (TCL)

TCL employs a range of methods to address data-related challenges, and it is important to note that there is no universally applicable algorithm that suits every situation. The choice of the right algorithm depends on several factors, including the particular problem at hand, the number of variables involved, the most suitable model for the task, and other pertinent factors. Below is a brief summary of some of the most commonly used machine learning algorithms.

**T1.　K-nearest neighbors (KNN):** The KNN algorithm is a type of nonparametric supervised machine learning method used for regression and classification. The method relies on the utilization of K training samples for predictions. In a study conducted by Chandel et al. in [73], the KNN method was applied to classify thyroid disease based on TSH, T4, and goiter parameters. Liu et al. [74] also employed the fuzzy KNN approach to differentiate between hyperthyroidism, hypothyroidism, and normal cases. There is a growing interest in larger datasets for future research, as noted in [75].

**T2.　Support vector machine (SVM):** An SVM is a machine learning method used for classification and regression tasks. In a study published in [76], an SVM approach was proposed for differentiating benign from malignant thyroid nodules by utilizing 98 thyroid nodule (TN) samples (82 benign and 16 malignant). Another study in [77] employed six SVMs to classify nodular thyroid lesions by selecting the most important textural characteristics. The authors reported that the proposed method achieved the correct classification. In [78], a generalized discriminant analysis and wavelet carrier vector machine system (GDA-WSVM) was introduced for diagnosing TN, consisting of feature extraction, classification, and testing phases.

**T3.　Decision trees (DT):** DT learning is a method for data mining that uses a predictive model for decision-making, where the output values are represented by the leaves and the input variables are represented by branches. This approach has been applied to uncover underlying thyroid diseases as demonstrated in various studies such as [79–82].

**T4.　Logistic regression (LR):** In [83], the LR model was used to determine the specific characteristics of thyroid microcarcinoma in 63 patients, based on the combination of contrast-enhanced ultrasound (CEUS) and conventional US values. Another study, conducted in northern Iran and reported in [84], applied LR to analyze 33,530 cases of thyroid cancer. LR is a widely used binomial regression model in machine learning.

### 2.4. Unsupervised Learning (USL)

In AI and computer science, USL involves analyzing data without pre-existing labels or annotations. It aims to uncover the underlying structures in the unlabeled data. Unlike supervised learning, which uses labeled data to calculate a success score, USL lacks this labeling, making it difficult to assess the accuracy of the results. While USL algorithms can perform more complex tasks compared to supervised ones, they can also be more unpredictable, adding unintended categories and introducing noise instead of structure. Despite these challenges, USL remains a valuable tool for exploring AI, as it enables the discovery of patterns and relationships in data that might not be immediately apparent [85,86].

Clustering (C)

The purpose of this method is to segment a set of thyroid cancer data into various homogeneous groups that possess similar characteristics, making it easier to classify the unlabeled datasets into benign and malignant. This detection approach has gained significant attention in various medical studies for its simplicity, including in the detection of DNA copy number changes [87], breast cancer recognition [88], cancer gene detection [89], skin cancer diagnosis [90], and brain tumor detection [91]. Additionally, clustering can also help identify cancer without precise definitions [92]. The clustering technique was used in [93] to identify factors that impact the normal functioning of TG, and DBSCAN and PCA were

applied to manage the clusters and reduce dimensionality. An automated clustering system for thyroid diagnosis was developed in [94] to prescribe the appropriate drug datasets for hyperthyroid, hypothyroid, and normal cases. The efficiency of fuzzy clustering for thyroid and liver datasets from the UCI repository was analyzed in [95], where the FPCM and PFCM algorithms were applied and compared.

**C1. K-means (KM):** The K-means (KM) method is a technique for data partitioning and a combinatorial optimization challenge. It is commonly utilized in USL, in which observations are separated into K groups. In [96], the authors explore the utilization of an ANN and improvised K-means method for normalizing raw data. The study used thyroid data from the UCI dataset containing 215 instances.

**C2. Entropy-based (EB):** In [97], a parameter-free calculation framework named DeMine was developed to predict microRNA regulatory modules (MRMs). DeMine is a three-step method based on information entropy. Firstly, the miRNA regulation network is transformed into a synergistic miRNA–miRNA network. Then, miRNA clusters are detected by maximizing the entropy density of the target cluster. Finally, the coregulated miRNAs are integrated into the corresponding clusters to form the final MRMs. The proposed method not only provides improved accuracy but also identifies more miRNAs as potential tumor markers for tumor diagnosis.

*2.5. Deep Learning (DL)*

DL is a subset of ML and AI that is based on ANNs with representation learning. ANNs are defined as a class of information processing systems comprised of interconnected nonlinear elements known as neurons. These networks have proven to be effective in addressing complex issues since they can store and retrieve information. An ANN with many hidden layers is commonly referred to as a DNN. The depth of the network allows it to capture increasingly abstract and high-level features as you progress through the layers. It can automatically learn, generate, and improve representations of data by employing large neural networks with many layers, hence the term "deep" learning. In thyroid cancer, DL has been deployed to perform different tasks, including: (i) Image classification—DL algorithms such as CNNs can be trained to classify thyroid ultrasound images. For instance, they can differentiate between benign and malignant nodules based on their shape, texture, and other characteristics [98–100]. This approach can significantly reduce the time and effort required for manual interpretation, thus aiding in the early detection and treatment of thyroid cancer. (ii) Pathological analysis—DL can also be utilized to analyze histopathological or cytopathological slide images, helping in the detection and classification of cancerous cells. (iii) Genomic data analysis—with the advent of genomic medicine, DL models can be employed to analyze genetic variations that may predispose individuals to thyroid cancer. (iv) Radiomics—DL models can be used to extract high-dimensional data from radiographic images, allowing for more precise and personalized treatment planning. (v) Predictive analysis—using electronic health records and other patient data, DL models can be used to predict the likelihood of a patient developing thyroid carcinoma, allowing for preventive measures to be taken if necessary. Figure 3 illustrates the different classifications of thyroid cancer using DL.

2.5.1. Extreme Learning Machine (ELM)

The ELM model features a layer of hidden nodes with a randomized weight distribution. The weights between the hidden node inputs and outputs are learned in a single step, resulting in a more efficient learning process compared to other models. The ELM has been proven to be an effective method in the diagnosis of TD, as evidenced in several studies such as [101–104].

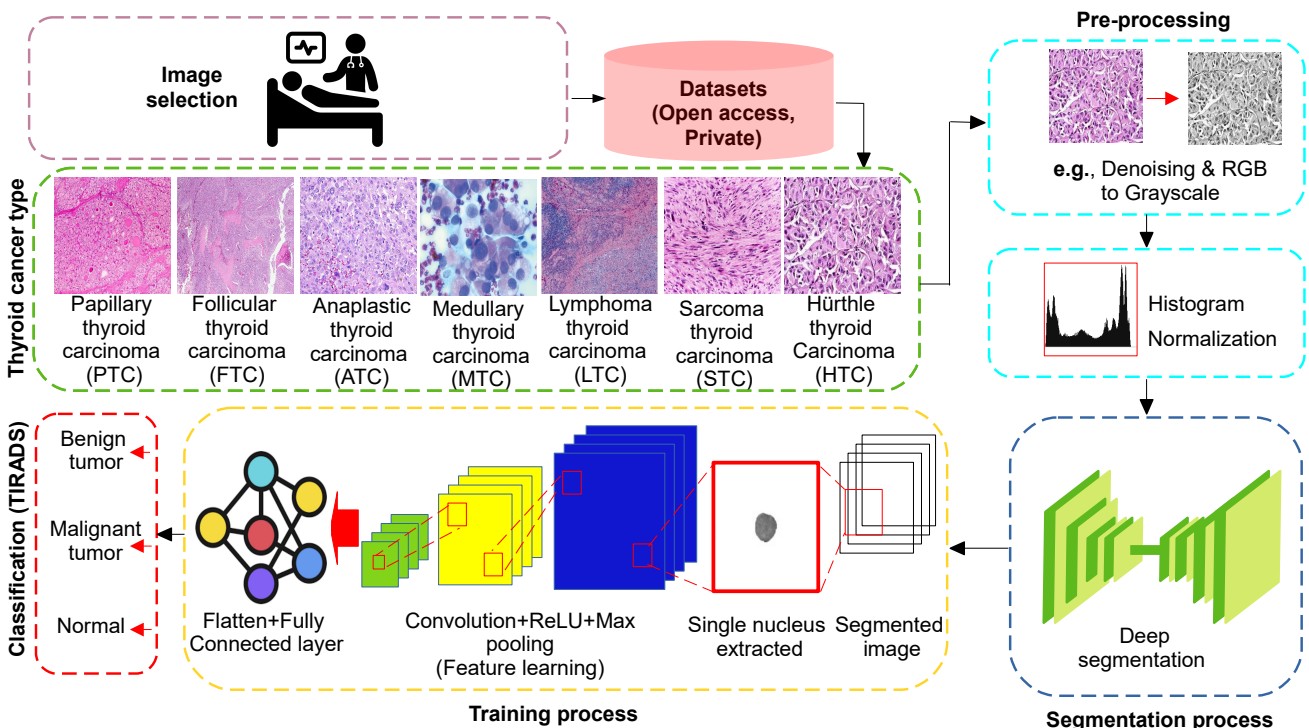

**Figure 3.** General DL system for thyroid cancer detection and classification.

### 2.5.2. Multilayer Perceptron (MLP)

MLP represents a category of feedforward networks where data are processed from the input layer through to the output layer. Each layer in this network comprises a varying number of neurons. Rao et al. [105] introduced an innovative approach for thyroid nodule classification, utilizing an MLP with a backpropagation learning algorithm. In their model, the MLP included four neurons in the input layer, three neurons in each of the ten hidden layers, and a single neuron in the output layer. Hosseinzadeh et al. [106] conducted a separate study with the objective of improving the accuracy of TD diagnosis through MLP networks. The research compared their findings with the existing literature on thyroid cancer classification and found MLP networks to be superior. Isa et al. [107] delved into the exploration of activation functions within MLP networks. Their goal was to identify the optimal activation function for the accurate classification of incurable diseases such as TD and breast cancer. The study evaluated multiple activation functions, including logarithmic, sigmoid, neural, sinusoidal, hyperbolic tangent, and exponential functions. The research found the hyperbolic tangent function to be the most effective for TD classification, using the backpropagation algorithm as the training algorithm. This result was further corroborated by Mourad et al. [108].

### 2.5.3. Radial Basis Function (RBF)

In [109], ML is applied to the classification of TN, where the MLP and RBF activation functions are utilized. The RBF activation function is found to outperform the MLP in terms of the structural classification of thyroid nodules. This approach highlights the effectiveness of activation functions in approximating functions, classifying, and predicting time series data, especially in the diagnosis of thyroid cancer.

### 2.5.4. Denoising Autoencoder (DAE)

DAEs can be beneficial for thyroid carcinoma classification by effectively learning representations from ultrasound or histopathological images. A DAE is a specific type of artificial neural network trained to reconstruct input data, often used for the purposes of dimensionality reduction or feature learning. The process for utilizing DAEs for thyroid carcinoma

classification generally follows these steps (i) preprocessing, (ii) noisy input creation, (iii) DAE training, (iv) feature extraction, and (v) classification. In [110], the authors implemented six autoencoder algorithms in the training process for PTC classification, including fixing weights and fine-tuning the network. The encoding layers and the complete autoencoder were used to embed the network. Another study [111] employed DAEs and stacked DAEs to extract features and identify informative genes in thyroid cancer.

### 2.5.5. Convolutional Neural Network (CNN)

CNNs are a class of DL models that have shown extraordinary performance in various image processing and analysis tasks, including the classification of medical images. CNNs are especially adept at processing gridlike data, such as an image, where spatial relationships between the pixels are crucial to understanding the image content. The past few years have seen considerable effort invested in developing CNN-based methodologies for detecting thyroid cancer, especially for the automated identification and classification of nodules in ultrasound imagery [112]. The ConvNet model, a widely adopted framework within the neural network realm, emphasizes the use of convolution operations over matrix multiplications [113]. Various CNN architectures such as LeNet [114], AlexNet [115], VGG [100], ResNet [116], GoogLeNet [117], Squeeze Net [118], and DenseNet [119] are distinguished by their incorporation of key components including convolutional, pooling, and fully connected layers.

In a study conducted by [120], the potential of CNN models to prognosticate thyroid cancer was explored using 131,731 ultrasound images taken from 17,627 thyroid cancer patients. Another research effort [121] employed VGG16, Inception, and Inception-Resnet models to differentiate malignant tissues within a set of 451 thyroid images from the DDTI dataset. To mitigate the challenge of data scarcity, the images were augmented before classification. A comparison of DCNN diagnostic performance with expert radiologists in distinguishing thyroid nodules within ultrasound images was carried out by [122], involving a test set of 15,375 TN ultrasound images. They utilized CNNE1 and CNNE2 models, derived from deep convolutional neural network (DCNN), for differentiating between malignant and benign TN. The study [123] proposed a CNN-based DL technique for detecting and classifying TN and breast nodules, with the results contrasted against those from ultrasound imaging. Table 2 presents a summary of recent CNN-based thyroid cancer classification contributions.

**Table 2.** Summary of CNN research conducted in the diagnosis of thyroid cancer. Accuracy, sensitivity, and specificity are provided in percentages (%) for better comparison.

| Ref. | Year | Country | NP | NM | NF | NN | NBN | NMN | TP | TN | FP | FN | ACC | Sens. | Spec. |
|------|------|---------|------|-----|------|------|------|------|------|-----|-----|------|-------|-------|-------|
| [124] | 2021 | China | 102 | 00 | 102 | 104 | 57 | 47 | 38 | 07 | 07 | 50 | 44.12 | 43.18 | 50.00 |
| [125] | 2021 | China | 102 | 25 | 77 | 103 | 73 | 33 | 27 | 12 | 06 | 61 | 36.79 | 30.68 | 66.67 |
| [126] | 2021 | Koria | 325 | 61 | 264 | 325 | 257 | 68 | 48 | 52 | 20 | 205 | 30.77 | 18.97 | 72.22 |
| [127] | 2021 | China | 2775 | 726 | 2049 | 2775 | 2472 | 303 | 271 | 363 | 32 | 2109 | 22.85 | 11.39 | 91.90 |
| [127] | 2021 | China | 163 | 48 | 115 | 175 | 67 | 108 | 86 | 09 | 22 | 58 | 54.29 | 59.72 | 29.03 |
| [128] | 2020 | China | 2489 | 614 | 1875 | 2489 | 1021 | 1468 | 1280 | 258 | 188 | 763 | 61.79 | 62.65 | 57.85 |
| [129] | 2020 | USA | 571 | 234 | 337 | 651 | 500 | 151 | 133 | 214 | 18 | 287 | 53.22 | 31.67 | 92.24 |
| [130] | 2020 | China | 166 | 46 | 100 | 209 | 109 | 100 | 87 | 16 | 13 | 93 | 49.28 | 48.33 | 55.17 |
| [122] | 2020 | Korea | 200 | 49 | 151 | 200 | 102 | 98 | 90 | 41 | 08 | 61 | 65.50 | 59.60 | 83.67 |
| [131] | 2020 | Korea | 340 | 79 | 261 | 348 | 252 | 96 | 31 | 25 | 65 | 227 | 16.09 | 12.02 | 27.78 |
| [132] | 2019 | Korea | 106 | 29 | 77 | 2018 | 132 | 86 | 69 | 23 | 17 | 109 | 42.20 | 38.76 | 57.50 |
| [133] | 2019 | China | 171 | 32 | 139 | 180 | 85 | 95 | 86 | 50 | 09 | 35 | 75.56 | 71.07 | 84.75 |

Abbreviations: number of patients (NP); number of males (NM); number of females (NF); number of nodules (NN); number of benign nodules (NBN); number of malignant nodules (NMN); true negative (TN); true positive (TP); false negative (FN); false positive (FP).

### 2.5.6. Recurrent Neural Network (RNN)

RNNs are a class of artificial neural networks where connections between nodes form a directed graph along a sequence, thus enabling them to use their internal state (memory) to process variable-length sequences of inputs. This unique feature makes RNN particularly suitable for tasks where temporal dependencies are essential, such as time-series analysis, language translation, and speech recognition. In the context of thyroid carcinoma classification, RNNs can be utilized to analyze sequential or time-dependent data, such as the development of a patient's clinical signs over time, the evolution of a tumor seen in a series of medical images, or changes in the gene expression related to the progression of thyroid cancer. For instance, in the study by Chen et al. (2017) [134], the authors propose a hierarchical RNN approach for classifying thyroid nodules based on historical ultrasound reports. This hierarchical RNN is composed of three layers, with each layer incorporating an individually trained long short-term memory (LSTM) network. The study's findings indicate that the hierarchical RNN model surpasses basic models in terms of computational efficiency, control accuracy, and robustness, making it an effective tool for diagnosing TN. These advantageous attributes stem from the inherent memory mechanisms of RNNs, which allow them to remember previous states through feedback loops. This memory capability renders RNNs a popular choice for applications in cancer detection.

### 2.5.7. Restricted Boltzmann Machine (RBM)

An RBM is a type of artificial neural network and a generative stochastic model. It was first introduced by Paul Smolensky [135] in 1986 under the name "Harmonium," but the concept of a "restricted" Boltzmann machine was developed by Geoffrey Hinton and his students in the mid-2000s. RBMs have a layer of visible units and a layer of hidden units, but no connections within layers—this is the restriction in their name. Each node in the layer is connected to every node in the other layer. The lack of intralayer connections simplifies the learning process. The work by Vairale et al. [136] presents an application of RBMs to develop a personalized fitness recommendation system tailored for individuals diagnosed with thyroid conditions. RBMs are a particular class of generative artificial neural networks characterized by a bidirectional architecture, which operates in an unsupervised manner. This structure comprises a visible layer containing binary variables and a hidden layer, also populated with interconnected binary variables. The learning process within RBMs is primarily conducted through a statistical analysis.

### 2.5.8. Generative Adversarial Network (GAN)

This type of ML network is composed of two distinct models: a generator and a discriminator. The generator maps a random input vector to an output in the data space, while the discriminator serves as a binary classifier that evaluates both input data from the training set and output data from the generator. GANs have gained widespread use in the diagnosis of diseases, including thyroid nodules [137,138].

### 2.5.9. Probabilistic Models (PM)

PMs, including Bayesian networks (BNs), are foundational concepts in computer science and statistics. These models represent uncertainty and dependencies among variables. In computer science, PMs are used for tasks such as ML, where they aid decision-making by quantifying uncertainty and modeling complex relationships. A BN, a specific type of PM, uses directed acyclic graphs to represent dependencies between variables. They are valuable for reasoning under uncertainty, making predictions, and handling incomplete information. In statistics, these models facilitate data analysis by incorporating probabilistic reasoning to draw meaningful inferences and estimate parameters, enhancing our understanding of complex systems. BNs have proven effective in the identification of various diseases. For instance, they have been employed to differentiate between benign and malignant thyroid nodules [139,140], as well as in the diagnosis of thyroid cancer, hepatitis, and breast cancer [141].

*2.6. Ensemble Methods (EMs)*

To address the complexity of cancer data and achieve higher accuracy in detection, the use of ensemble methods is commonly employed in the field. An ensemble method involves dividing the data into subgroups and applying multiple machine learning techniques to each subgroup simultaneously, then synthesizing the results to make a final diagnosis. By combining multiple models, the ensemble method aims to produce an optimal predictive model for thyroid cancer detection. This approach has been shown to be effective in various studies, such as [142], where the authors emphasize the importance of ensemble methods in achieving a more comprehensive understanding of the data and improving the accuracy of the diagnosis.

2.6.1. Bagging (B)

In the realm of thyroid cancer screening, bagging is an ensemble learning technique utilized to improve the accuracy and stability of ML algorithms. This algorithm operates by reducing variance and avoiding overfitting and can be applied to a variety of methods, particularly decision trees. The purpose of bagging is to enhance the performance of weak classifiers in the field of thyroid cancer screening applications.

**B1. Bootstrap aggregation (BA):** The bootstrap aggregating technique is a widely utilized ensemble method aimed at improving the accuracy of machine learning algorithms, particularly for the purposes of classification, regression, and variance reduction. In [143], this approach was employed for diagnosing thyroid abnormalities.

**B2. Feature bagging (FB):** In [144], FB is introduced as a method of ensemble learning with the goal of minimizing the correlation between the individual models in the ensemble. FB achieves this by training the models on a randomly selected subset of features, instead of all features in the dataset. The method was applied to distinguish between benign and malignant thyroid cancer cases [145].

2.6.2. Boosting (O)

Meta-algorithms are often used in USL to mitigate the variance and enhance the performance of weak classifiers by transforming them into strong classifiers.

**O1. AdaBoost** In the study by Pan et al. [146], a new method called AdaBoost was utilized to diagnose thyroid nodules using the standard UCI dataset. The RF and PCA techniques were employed for classification purposes and to maintain data variability, respectively.

**O2. Gradient tree boosting (XGBoost):** In [147], the XGBoost algorithm was introduced as a fast and efficient implementation of gradient-boosted decision trees. Since its introduction, the XGBoost algorithm has been applied to a range of research topics, including civil engineering [148], time-series classification [149], sport and health monitoring [150], and ischemic stroke readmission [151].

For thyroid cancer detection, the authors in [152] used XGBoost to diagnose benign and malignant thyroid nodules, as a solution to the challenge of obtaining accurate diagnoses with DL models when a large-scale dataset is unavailable.

Table 3 provides a summary of research frameworks for the detection of benign and malignant thyroid cancers, including the category, classifier, detected disease, dataset, objective, and used quantifiable metrics. This table helps to categorize AI methods used for thyroid cancer detection and highlights the current key applications.

**Table 3.** Summary of research frameworks conducted in the detection of benign and malignant thyroid cancer.

| Ref. | Category | Classifier | DD | Dataset | O | SV | APP |
|------|----------|-----------|-----|---------|---|-----|-----|
| [110] | DL | DAE | PTC | TCGA | O1 | 18,985 features | US |
| [111] | DL | DAE | PTC | TCGA | O1 | 510 samples | Omics |
| [67] | DL | CNN | TC | PD | O1 | 10,068 images | Omics |
| [153] | DL | CNN | TC | PD | O1 | 482 images | Omics |
| [154] | DL | CNN | PTC, FTC | NA | NA | NA | FNAB |
| [155] | DL | CNN | PTC | PD | O1 | 370 microphotographs | FNAB |
| [156] | DL | CNN | PTC | PD | O3 | 469 patients | FNAB |
| [157] | DL | CNN | TC | DDTI | O1 | 298 patients | US |
| [158] | DL | CNN | TC | PD | O1 | 1037 images | US |
| [159] | DL | CNN | TN | PD | O2 | 80 patients | US |
| [160] | DL | CNN | TN | PD | O2 | 300 images | US |
| [161] | DL | CNN | TC | PD | O1 | 459 labeled | US |
| [162] | DL | CNN | TD | ImageNet | O1 | 2888 samples | US |
| [120] | DL | CNN | TC | PD | O1 | 17,627 patients | US |
| [121] | DL | CNN | TC | PD | O1 | 1110 images | US |
| [123] | DL | CNN | TN | PD | O1, S1 | 537 images | US |
| [134] | DL | RNN | TN | PD | O1 | 13,592 patients | US |
| [136] | DL | DBM | TD | PD | O1 | 94 users | Fitness |
| [138] | DL | GAN | TC | PD | O3 | 109 images | Surgery |
| [163] | DL | NA | TC | NA | NA | NA | US |
| [164] | DL | NA | TC | PD | O1 | 1358 images | US |
| [165] | AI | NA | TC | PD | O1 | 50 patients | Surgery |
| [166] | AI | NA | TC | PD | O1 | 89 patients | US |
| [101] | ANN | ELM | TD | UCI | O1 | 215 patients | US |
| [102] | ANN | ELM | TD | UCI | O1 | 215 patients | US |
| [103] | ANN | ELM | TD | PD | O1 | 187 patients | US |
| [105] | ANN | MLP | TD | PD | O1 | 7200 samples | US |
| [106] | ANN | MLP | TD | UCI | O1 | 7200 patients | US |
| [109] | ANN | RBF | TD | PD | O1 | 487 patients | US |
| [167] | ANN | RBF | TD | PD | O1 | 447 patients | Cytopathological |
| [168] | ANN | NA | FTC | PD | O1 | 57 smears | FNAB |
| [169] | ANN | NA | FTC | NA | NA | NA | FNAB |
| [170] | ANN | NA | TC | TCGA | O3 | 482 samples | Histopathological |
| [171] | ANN | NA | TC | PD | O1 | 1264 patients | FNAB |
| [172] | ANN | NA | TN | PD | O1 | 276 patients | US |
| [73] | TCL | KNN | TD | PD | O1 | 7200 instances | US |
| [173] | TCL | KNN | FTC | PD | O1, O2 | 94 patients | Histopathological |
| [174] | TCL | SVM | FTC | PD | O1 | 43 nuclei | Histopathological |
| [175] | TCL | SVM | TN | PD | O1 | 467 TN | US |
| [76] | TCL | SVM | TC | PD | O1 | 92 subjects | US |
| [176] | TCL | SVM | PTC | TCGA | O1 | 500 patients | Omics |
| [177] | DL | DL | PTC | TCGA | O3 | 115 slides | Omics |
| [178] | ML | ML | TN | PD | O1 | 121 patients | Omics |
| [79] | TCL | DT | TC | UCI | O1 | 3739 patients | US |
| [81] | TCL | DT | TC | NA | O1 | NA | US |
| [82] | TCL | DT | TC | UCI | O1 | 499 patients | US |
| [83] | TCL | LR | TC | PD | O1 | 63 patients | US |
| [84] | TCL | LR | TN | PD | O1 | 33,530 patients | US |
| [139] | PM | BN | TD | UCI | O1 | 93 adult patients | US |
| [140] | PM | BN | TC | NA | O1 | 37 patients | US |
| [96] | C | KM | TC | UCI | O1 | 215 instances | US |
| [97] | C | EB | TC | Private data | O1 | 734 cases | US |
| [70] | DR | PCA | TC | PD | O1 | NA | NA |
| [144] | B | FB | TN | PD | O1 | 1480 patients | US |

Abbreviation: application (APP), detected disease (DD), objective (O), thyroid cancer (TC), subjects for validation (SV), private data (PD).

## 3. Thyroid Cancer Datasets

In the field of thyroid carcinoma research, a number of datasets have been developed to facilitate the validation of ML algorithms and models. This is especially important because the creation of such datasets is a major challenge in the area of endocrine ML. In this section, we present an overview of the most significant thyroid databases, which offer a set of standards for evaluating the performance of learning methods and assist in the diagnosis and monitoring of complicated diseases.

- **ThyroidOmics:** This is a dataset developed by the Thyroid Working Group of the CHARGE Consortium that aims to examine the underlying factors and consequences of TD using various omics techniques such as genomics, epigenomics, transcriptomics, proteomics, and metabolomics. The dataset consists of the results of the discovery stage of the genomewide association analysis (GWAS) meta-analysis for thyrotropin (TSH), free thyroxine (FT4), increased TSH (hypothyroidism), and decreased TSH (hyperthyroidism) as reported in [179,180].
- **Thyroid Disease Data Set (TDDS):** The dataset utilized for classifying using ANN is referred to as the Thyroid database and features 3772 training instances and 3428 testing instances, with a combination of 15 categorical and 6 real attributes. The three defined classes in this dataset include normal (not hypothyroid), hyperfunctioning, and subnormal functioning [181].
- **KEEL Thyroid Dataset:** The KEEL dataset provides a set of benchmarks to evaluate the effectiveness of various learning methods. This dataset includes several types of classification, such as standard, multi-instance, imbalanced data, semi-supervised classification, regression, time series, and USL, which can be used as reference points for a performance analysis [182].
- **TNM8 Dataset:** A dataset was created for the purpose of reporting pathologies of thyroid resection specimens associated with carcinoma. The data do not include core needle biopsy specimens or metastasis to the thyroid gland. The dataset also does not encompass noninvasive follicular thyroid neoplasm with papillary-like nuclear features (NIFTP), tumors of uncertain malignancy, thyroid carcinomas originating from struma ovarii, carcinomas originating in thyroglossal duct cysts, sarcomas, or lymphomas.
- **Gene Expression Omnibus (GEO):** The GEO database is a genomics repository that follows the guidelines of the minimum information about a microarray experiment. This database is designed to store gene expression datasets, arrays, and sequences and provides researchers with access to a vast collection of experiment results, gene profiles, and platform records in GEO [183].
- **Surveillance, Epidemiology, and End Results (SEER):** The creators of this dataset aim to supply a collection of clinical characteristics from thyroid carcinoma patients, which includes 34 details such as age, gender, lymph nodes, and others.
- **Digital Database Thyroid Image (DDTI):** The DDTI dataset serves as a valuable resource for researchers and new radiologists looking to develop algorithm-based CAD systems for thyroid nodule analysis. The dataset comprises 99 cases and 134 images, with each patient's data stored in an XML file format [184]. Figure 4 provides an illustration of six samples from each of the thyroid carcinoma tissue types in the DDTI dataset.
- **Cancer Genome Atlas (TCGA):** The TCGA is a comprehensive collection of data gathered from 11,000 patients diagnosed with various types of cancer over a period of 12 years. The data consist of detailed genomic, epigenomic, transcriptomic, and proteomic information, amounting to a total of 2.5 petabytes. This extensive dataset has been instrumental in advancing the research, diagnosis, and treatment of cancer.
- **National Cancer Data Repository (NCDR):** The NCDR serves as a resource for healthcare and research with the goal of capturing all recorded cases of cancer in England. These data are sourced from the office for national statistics [185].
- **Prostate, Lung, Colorectal, and Ovarian (PLCO) dataset** The National Cancer Institute supports the PLCO cancer screening trial, aimed at examining the direct factors that contribute to cancer in both men and women. The trial has records of 155,000 participants, and all studies regarding thyroid cancer incidence and mortality can be found within it [186].

In Table 4, we present examples of public and private TCDs used in thyroid cancer detection.

**Table 4.** Examples of public and private TCDs used in thyroid cancer detection.

| Ref | Year | TCD | IT | IF | Instance | M/F | DA |
|---|---|---|---|---|---|---|---|
| [44] | 2018 | BMU | Sonographic | PNG | 1077 | 4309 | Public |
| [172] | 2019 | TCCC | US | PNG | 370 | 370 | Public |
| [187] | 2019 | Clinical | US | JPEG | 117 | 2108 | Public |
| [188] | 2019 | Hospital | US | JPEG | 62 | 12/60 | Public |
| [189] | 2020 | TIRADS | US | JPEG | 5278 | NA | Public |
| [190] | 2018 | Peking Union | US | JPEG | 4309 | 1179 | Private |
| [120] | 2019 | Medical Center | US | PNG | 1425 | 2064 | Private |
| [191] | 2020 | PubMed | CT scans | JPEG | 2108 | 54/253 | Private |
| [156] | 2021 | ACR | DICOM | DICOM | 1629 | 83/289 | Private |
| [126] | 2021 | Clinical | US | PNG | 40 | 407 | Private |

Abbreviations: image types (IT); image format (IF); dataset access (DA); male (M); female (F).

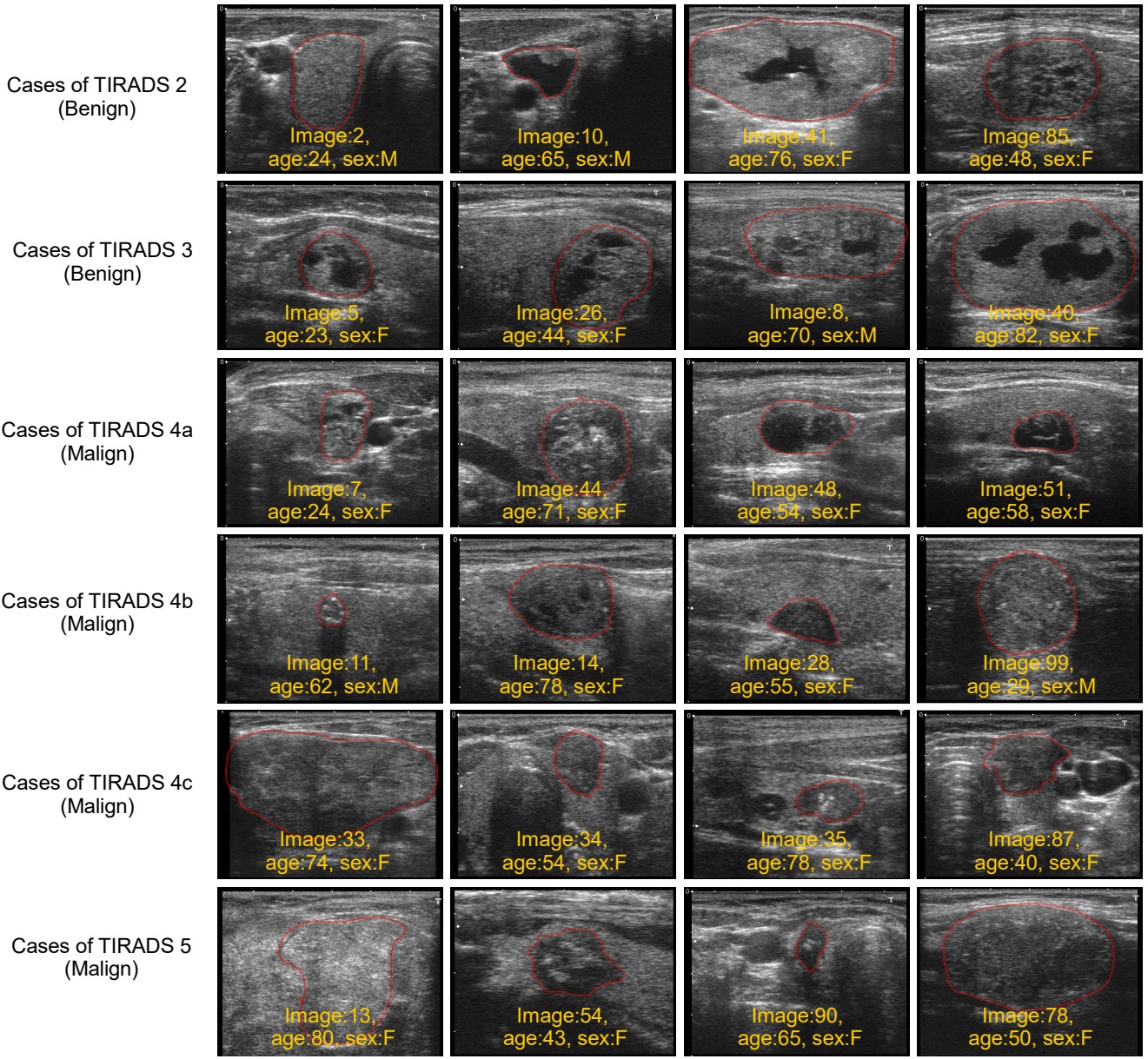

**Figure 4.** Example of six samples for each class from the DDTI datasets.

The strengths and weaknesses of AI-based thyroid cancer detection techniques are summarized in Table 5.

**Table 5.** A summary of thyroid cancer detection techniques AI-based, including their strengths and weaknesses.

| Ref. | AI Meth. | Achieve. (%) | Advantages | Drawbacks |
|------|----------|--------------|------------|-----------|
| [192] | DAE | ACC = 92.9 | No need for labels for thyroid cancer | Insufficient training data and need relevant data |
| [193] | CNN | AUC = 85.0 | High thyroid cancer detection | Insufficient labels for thyroid cancer and weak in interpretability |
| [194] | RNN | ACC = 98.2 | No need for labels for thyroid cancer | Slow computation and difficulty in training |
| [195] | MLP | ACC = 95.0 | Adaptive learning for thyroid cancer | Limited in its results |
| [196] | KNN | P = 93.0 | High sensitivity to thyroid cancer detection | Insufficient labels for thyroid cancer |
| [197] | SVM | ACC = 97.0 | High sensitivity to thyroid cancer detection | Weak in interpretability and long training time |
| [198] | DT | AUC = 73.10 | Does not require scaling and normalization of data | Unstable |
| [199] | LR | – | Low-cost training and easier implementation | Difficulty to label data |
| [200] | B | ACC = 94.88 | High detection of thyroid cancer | Loss of interpretability and high computational cost |

Abbreviations: area under curve (AUC).

## 4. Features

In this section, the focus is on showcasing the crucial techniques utilized in the classification process for characteristic extraction and selection. This primarily involves identifying a subset of relevant features that positively impact the classification accuracy and eliminating irrelevant variables.

### 4.1. Feature Selection Methods (FS)

**FS1. Information gain (IG):** Information gain (IG) is a straightforward method for classifying thyroid cancer features. This method evaluates the likelihood of having cancer by comparing the entropy before and after the examination. Typically, a higher gain value corresponds to a lower entropy. IG has been used extensively in several applications for the diagnosis of cancerous diseases, such as in filtering informative genes for precise cancer classification [201], selecting breast cancer treatment factors based on the entropy formula [202], analyzing and classifying medical data of breast cancer [203], reducing the dimensionality of genes in multiclass cancer microarray gene expression datasets [204], and filtering irrelevant and redundant genes of cancer [201]. In [205], IG is utilized as a feature selection technique to eliminate redundant and irrelevant symptoms in datasets related to diabetes, breast cancer, and heart disease. Additionally, the IG-SVM approach, combining IG and SVM, has been employed and its results served as input for the LIBSVM classifier [201].

**FS2. Correlation-based feature selection (CFS):** CFS is a technique frequently used for evaluating the correlation between different cancer features. In various studies, the CFS algorithm has been integrated into attribute selection methods for improved classification, such as in [206], where it was applied to thyroid, hepatitis, and breast cancer data from the UCI ML repository. In [141], the authors proposed a hybrid method that combined learning algorithm tools and feature selection techniques for disease diagnosis. The CFS was utilized in [207] for feature selection in microarray datasets to minimize the data's dimensionality and identify discriminatory genes. A hybrid model incorporating the CFS and a binary particle swarm optimization (BPSO) was proposed in [208] to classify cancer types and was applied to 11 benchmark microarray datasets. The CSVM-RFE, which involves a CFS, was used in [209] to reduce the number of cancer features and eliminate irrelevant ones. In [176], the authors utilized CFS techniques to identify key RNA expression features.

**FS3. Relief (R):** The relief algorithm, commonly known as RA, is an effective method used in selecting important features by assessing their differentiation quality by assigning scores. This technique calculates the weight of various features based on the correlation between cancer attributes. In a study published in [210], a feature selection method based on the relief algorithm was proposed as a means of improving efficiency.

**FS4. Consistency-based subset evaluation (CSE):** The study in [211] presents a hybrid classification model for breast cancer, which is based on dividing cancer data into single-

class subsets. The effectiveness of the model is evaluated using the Wisconsin Breast Cancer Dataset (WBCD).

*4.2. Feature Extraction Methods (FE)*

**FE1.  Principal components analysis (PCA):** The use of PCA has been highlighted in several studies as a method to reduce the dimensionality of data and decorrelate the attributes of cancer features.  For instance, in [69], PCA was applied to the dual-tree complex wavelet (DTCW) transform to select the optimum features of thyroid cancer. In [70], PCA was proposed as a tool for classifying different thyroid cancer subtypes such as papillary, follicular, and undifferentiated. The implementation of PCA and linear discriminant analysis was also explored in [212] for classifying Raman spectra of different thyroid cancer subtypes. Finally, in [213], the authors utilized PCA on cDNA microarray data to uncover the biological basis of breast cancers.

**FE2. Texture description:** Texture analysis is a commonly used method for extracting relevant information in the classification, segmentation, and prediction of thyroid cancer. There are numerous texture analysis techniques in the literature, including wavelet transform, binary descriptors, and statistical descriptors. The discrete wavelet transform (DWT), in particular, has received significant attention for its ability to perfectly decorrelate data. Many studies have utilized wavelets for thyroid cancer detection, such as in [214], where wavelet techniques were employed to identify cancer regions in thyroid, breast, ovarian, and prostate tumors. In [215], texture information was used to diagnose TN malignancy through a two-level 2D wavelet transform. Other works exploring this area can be found in [216,217].

**FE3. Active contour (AC):** The active contour (AC), first introduced by Kass and Witkin in 1987, is a dynamic structure primarily used in image processing.  There are several approaches for solving the problem of contour segmentation using a deformable curve model, which has seen numerous applications in the field of detection of thyroid cancer, as demonstrated in [218–220].

**FE4. Local binary patterns (LBP):** The LBP are features employed in computer vision to recognize textures or objects in digital images.  LBP have been utilized to detect thyroid cancer in [216]. The combination of LBP and DL has also been proposed to classify benign and malignant thyroid nodules in [221,222].

**FE5. Gray-level co-occurrence matrix (GLCM):** The GLCM is a matrix that represents the distribution of values of pixels that occur together at a specified offset in an image. In [223], GLCM was used to extract features to differentiate between different types of thyroid cancer. In [224], the differences between an individual with Hashimoto's thyroiditis-associated PTC and one with Hashimoto's thyroiditis alone were investigated based on GLCM comparison.

**FE6. Independent component analysis (ICA):** In an ICA, information is gathered into a set of contributing features for the purpose of feature extraction. ICA is utilized to separate multivariate signals into their individual components.  In [225], ICA is used to extract 29 attributes as independent and useful features for classifying data into either hypothyroid or hyperthyroid using an SVM.

A summary of feature selection and extraction methods based on DL conducted in the diagnosis of thyroid cancer are illustrated in Table 6.

**Table 6.** Summary of feature selection and extraction methods based on DL conducted in the diagnosis of thyroid cancer.

| Ref. | Year | Classifier | Features | Contributions |
|---|---|---|---|---|
| [226] | 2017 | KNN | FC/IG | - Avoids data redundancy and reduces computation time. The KNN algorithm deals with the missing data, and the ANFIS algorithm is provided with the resultant data as input. |
| [227] | 2017 | SVM | FC/CFS | - Extracts the geometric and moment features while some kernels of the SVM classifier classify the extracted features. |
| [108] | 2020 | CNN | FC/R | - Combines ML and feature selection algorithms (namely, Fisher's discriminant ratio, Kruskal–Wallis' analysis, and Relief-F) to analyze the SEER database. |
| [228] | 2022 | CNN | FE/PCA | - The influence of unbalanced serum Raman data on the prediction results was minimized by using an oversampling algorithm in this study. PCA was used to reduce the data dimension before classifying the data using RF and adaptive boosting. |
| [229] | 2012 | O | FE/TD | - Combines CAD and DWT and texture feature extraction. The AdaBoost classifier uses the extracted features to classify images into benign or malignant thyroid images. |
| [230] | 2021 | CNN | FE/AC | - Image enhancement, segmentation, and multifeature extraction, encompassing both geometric and texture features. Each characteristic is then classified using an MLP and SVM, resulting in a determination of either benign or malignant. |
| [189] | 2020 | SVM | FE/LBP | - Deep features are extracted by a CNN and are combined with handcrafted features, including a histogram of oriented gradient (HOG), and scale-invariant feature transform to create fused features. These fused features are then used for classification by an SVM. |
| [231] | 2019 | SVM | FE/GLCM | - Uses a median filter to reduce noise and delineates the contours before extracting features from thyroid regions, including GLCM texture features. SVM, RF, and bootstrap aggregating (bagging) are then used to identify the benign and malignant nodules. |
| [225] | 2019 | SVM | FE/ICA | - A multikernel-based SVM is used as a classifier to distinguish the thyroid disease. |

## 5. Standard Assessment Criteria

In this section, we examine the most commonly utilized standard parameters for evaluating the identification of TD. These criteria serve as a measure of the effectiveness of the methods used. Selecting the right metric is crucial when evaluating the performance of machine learning models. Numerous metrics have been proposed to evaluate machine learning models in various applications. Here, we present a summary of popular metrics that are considered suitable for assessing the performance of AI algorithms applied in the detection of thyroid cancer (See Tables 7–9).

*Classification and Regression Metrics*

**Table 7.** Summary of classification and regression metrics used in evaluating AI-based thyroid cancer detection schemes.

| Metric | Mathematical Formula | Description |
|---|---|---|
| Accuracy (ACC) | $\frac{T_P + T_N}{T_P + F_P + T_N + F_N} 100\%$ | Gives the correct percent of the total number of positive and negative predictions. |
| Specificity | $\frac{T_N}{T_N + F_P} 100\%$ | It is the ratio of correctly predicted negative samples to the total negative samples. |
| Sensitivity | $\frac{T_P}{T_P + F_N} 100\%$ | It is a quantifiable measure metric of real positive cases that were predicted as true positive cases. |
| Precision (P) | $\frac{T_P}{T_P + F_P} 100\%$ | Measures the proportion of true positive predictions made by the model, out of all the positive predictions made by the model. |
| F1 score (F1) | $2 \times \frac{Precision \times Recall}{Precision + Recall}$ | It is the harmonic mean of precision and sensitivity of the classification. |
| Error rate (ER) | $\frac{F_N + F_P}{T_P + F_N + F_P + T_N} 100\%$ | It is equivalent to one minus accuracy. |
| Root-mean-square error (RMSE) | $\left( \sqrt{1 - (ER)^2} \right) \times SD$ | It is the standard deviation of the predicted error between the training and testing dataset, its lower value means that the classifier is an excellent one. |
| The negative predictive value (NPV) | $\frac{T_N}{T_N + F_N}$ | It is the proportion of negative results in diagnostic tests; a higher value means the accuracy of the diagnosis. |
| Jaccard similarity index (JSI) | $\frac{|A \cap B|}{|A \cup B|} = \frac{T_P}{T_P + F_P + F_N}$ | It has been proposed by Paul Jaccard to gauge the similarity and variety in samples. |
| Fallout or false positive rate (FPR) | $\frac{F_P}{F_P + T_N} = 1 - SP$ | Measures the proportion of negative samples that are incorrectly classified as positive by the model. |
| Volumetric overlap error (VOE) | $\frac{F_P + F_N}{T_P + F_P + F_N}$ | Evaluates the similarity between the segmented region and the ground-truth region. VOE measures the amount of overlap between the two regions and is defined as the ratio of the volume of the union of the segmented region and the ground-truth region to the volume of their intersection. |
| Mean absolute error (MAE) | $\frac{1}{N} \sum_{i=1}^{N} |a_i - p_i|$ | It is the average of the difference between the original values and the predicted values. |
| Mean squared error (MSE) | $\frac{1}{N} \sum_{i=1}^{N} (y_i - r_i)^2$ | It is the average of the square of the difference between the original values and the predicted values. |

*Statistical Metrics*

**Table 8.** Summary of statistical metrics used in assessing AI-based thyroid cancer detection schemes.

| Metric | Mathematical Formula | Description |
|---|---|---|
| Standard deviation (SD) | $\sqrt{\sum(x-\mu)^2/N}$ | It is a measure of the amount of variation or dispersion in a set of data. |
| Correlation (Corr) | $(\sum((x-\mu x)\cdot(y-\mu y)))/(\sqrt{(\sum(x-\mu x)^2)}\cdot\sqrt{(\sum(y-\mu y)^2)})$ | It describes the degree of association or relationship between two or more variables. |
| Kappa de Cohen | $k=\frac{\Pr(a)-\Pr(e)}{1-\Pr(e)}$ | It measures the degree of concordance between two evaluators, relative to chance. |

*Computer Vision Metrics*

**Table 9.** Summary of computer vision metrics used in assessing AI-based thyroid cancer detection schemes.

| Metric | Mathematical Formula | Description |
|---|---|---|
| Peak signal-to-noise ratio (PSNR) | $10\cdot\log_{10}((MAX_I^2)/MSE)$ | It measures the ratio of the maximum possible power of a signal to the power of the noise that affects the fidelity of its representation. |
| Structural similarity index (SSIM) | $MSSIM(x,y)=\frac{1}{L}\sum_{i=1}^{L}SSIM(x_i,y_i)$ | It evaluates the similarity between two images or videos by comparing their luminance, contrast, and structural information. |
| Visual information fidelity (VIF) | $\frac{\sum_j I(C^j;F^j/s^j)}{\sum_j I(C^j;E^j/s^j)}$ | It evaluates the quality of a reconstructed or compressed image or video compared to the original signal. It measures the amount of visual information preserved in the processed image or video, taking into account the spatial and frequency characteristics of the image. |
| Normalized cross-correlation (NCC) | $\frac{\sum_{i=1}^{M}\sum_{j=1}^{N}(I(i,j)-R(i,j))^2}{\sum_{i=1}^{M}\sum_{i=1}^{N}I(i,j)^2}$ | Measures the similarity between two images (or videos) by subtracting the mean value of each signal from the signal itself. Then, the signals are normalized by dividing them by their standard deviation. Finally, the cross-correlation between the two normalized signals is calculated. |
| Structural content (SC) | $\frac{\sum_{i=1}^{M}\sum_{j=1}^{N}I(i,j)^2}{\sum_{i=1}^{M}\sum_{j=1}^{N}R(i,j)^2}$ | A higher value of structural content shows that the image is of poor quality. |
| Weight PSNR | $10\log\left(\frac{(2^n-1)^2}{NVF\times MSE}\right)$ | It takes into account the image texture [232]. |
| Noise visibility function (NVF) | $\text{Normalization}\left\{\frac{1}{1+\delta_{bloc}^2}\right\}$ | It estimates the texture content in the image. $\delta_{bloc}$ is the luminance variance. |
| Visual signal-to-noise ratio (VSNR) | $10\log_{10}\left(\frac{C^2(I)}{(VD)^2}\right)$ | It is based on the specified thresholds of distortions in the image based on the computing of contrast thresholds and a wavelet transform. If the distortions are lower than the threshold, the VSNR is perfect. $C(I)$ is the RMS contrast of the original image $I$, and $VD$ is the visual distortion [233]. |
| Weighted signal-to-noise ratio (WSNR) | $10\log_{10}\left(\frac{\sum_{u=0}^{M-1}\sum_{v=0}^{N-1}|A(u,v)C(u,v)|^2}{\sum_{u=0}^{M-1}\sum_{v=0}^{N-1}|A(u,v)-B(u,v)C(u,v)|^2}\right)$ | It is based on the contrast sensitivity function (CSF). $A(u,v)$, $B(u,v)$, and $C(u,v)$ represent discrete Fourier transforms (2D TFD) [234]. |
| Normalized absolute error (NAE): | $\frac{\sum_{i=1}^{M}\sum_{j=1}^{N}|I(i,j)-R(i,j)|}{\sum_{i=1}^{M}\sum_{j=1}^{N}|I(i,j)|}$ | It evaluates the accuracy of an ML model's predictions. It measures the difference between the predicted values and the actual values, as a proportion of the range of the actual values. |
| Laplacian mean squared error (LMSE) | $\frac{\sum_{i=1}^{M}\sum_{j=1}^{N}[L(I(i,j))-L(R(i,j))]^2}{\sum_{i=1}^{M}\sum_{j=1}^{N}[L(I(i,j))]^2}$ | It is a variant of the mean squared error ()! ()!) that uses the Laplacian distribution instead of the Gaussian distribution. $L(I(i,j))$ is the Laplacian operator. |

*Ranking Metrics*

**M1. Mean reciprocal rank (MRR):** The MRR is a statistic measure for evaluating the mean reciprocal rank of results for a sample of queries [235].

$$MRR=\frac{1}{|Q|}\sum_{i=1}^{|Q|}\frac{1}{rank_i} \qquad (1)$$

where $rank_i$ refers to the rank position of the first relevant document for the *i*th query.

**M2. The discounted cumulative gain (DCG):** the DCG is used to measure the ranking quality [236].

## 6. Example of Thyroid Cancer Detection Using AI

To explain how thyroid cancer has been considered in the literature and how AI can be used to detect types of cancers, in the following, we present a simple example of TD classification. It has been known that pattern recognition is the process of training a neural network to assign the correct target classes to a set of input patterns. Once trained, the network can be used to classify patterns. In this section, we present an example of thyroid cancer classification as benign, malignant, and normal based on a set of features specified according to the TIRADS. In this example, the dataset (7200 samples) was selected from the UCI Machine Learning Repository [237]. This dataset can be used to create a neural network that classifies patients referred to a clinic as normal, hyperfunctioning, or subnormal functioning. The thyroid inputs and thyroid targets are defined as: (i) TI: a $21 \times 7200$ matrix consisting of 7200 patients characterized by 15 binary and 6 continuous patient attributes. (ii) TT: a $3 \times 7200$ matrix of 7200 associated class vectors defining which of three classes each input is assigned to. Classes are represented by a one in rows 1, 2, or 3. (1) Normal, not hyperthyroid. (2) Hyperfunctioning. (3) Subnormal functioning.

In this network, the data were divided into 5040 samples, 1080 samples, and 1080 samples used for training, validation, and testing, respectively. The network was trained to reduce the error between thyroid inputs and thyroid targets or until it reached the target goal. If the ER did not decrease and the training did not improve, the training data were halted with the data of the validation set. The testing dataset was used to deduce the values of the targets. Thus, it determined the percentage of learning. For this example, 10 neurons were used in the hidden layer in this model for 21 inputs and 3 outputs. After the simulation of the model, the percent error was 5.337%, 7.407%, and 5.092% for training, validation, and testing, respectively. Thus, in total, it recognized 94.4% and the overall ER was 5.6%. The confusion matrix and the ROC metric are illustrated in Figure 5.

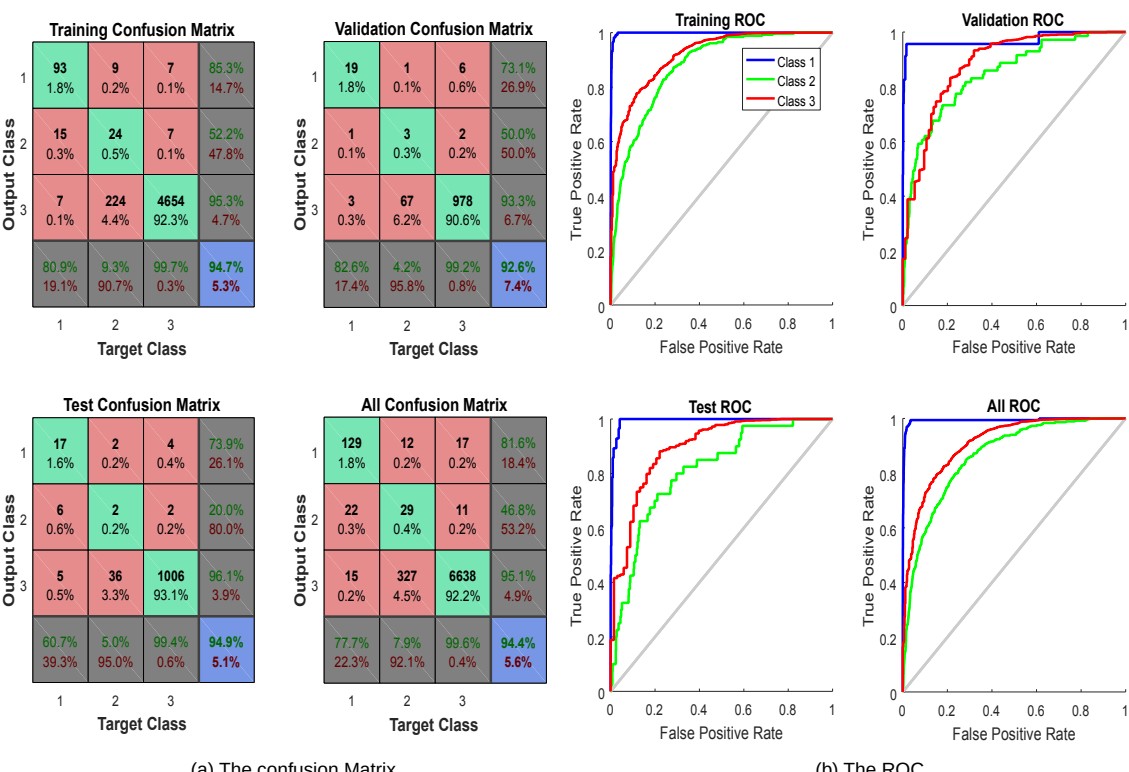

(a) The confusion Matrix                    (b) The ROC

**Figure 5.** An example of the confusion matrix and ROC metric for thyroid cancer classification.

Figure 6 illustrates an example of a thyroid segmentation in ultrasound images using K-means (three clusters were chosen for this example) which is one of the most commonly used clustering techniques.

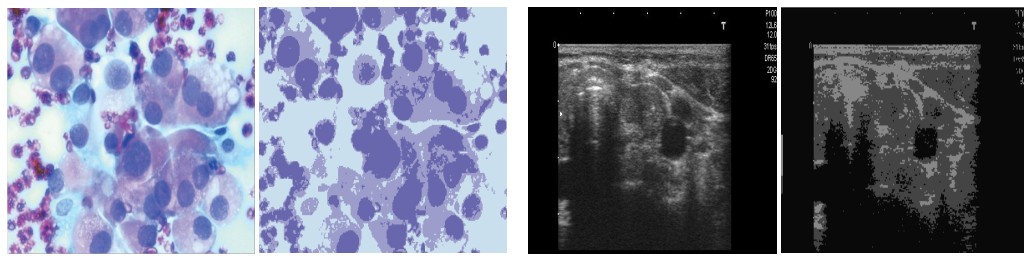

(a) Medullary thyroid carcinoma image　　　　　　　　(b) Thyroid ultrasound image

**Figure 6.** Example of thyroid segmentation based on the K-means method.

## 7. Critical Analysis and Discussion

As we delve into the core of this paper, it is essential to critically assess and discuss the multitude of facets associated with the application of AI in thyroid carcinoma detection. While the promise of AI has been well articulated in the existing literature, a more nuanced perspective is needed to fully understand its impact on healthcare, both positive and negative. In this section, we undertake a critical analysis of the effectiveness of AI models for thyroid carcinoma detection. Moving beyond the optimistic numbers, we question the robustness of these models in real-world clinical settings and discuss their role in the broader context of clinical decision-making. Furthermore, we explore the potential biases in AI models, understanding how they might inadvertently perpetuate existing inequities in healthcare. A comparative assessment of AI-based and traditional diagnostic methods provides deeper insights into their relative effectiveness. Moreover, acknowledging the challenges to the implementation of AI tools in healthcare, we delve into the infrastructural, regulatory, and cultural barriers that might hinder their widespread adoption. Lastly, we underscore the crucial role of interdisciplinary collaboration in ensuring the successful integration of AI into healthcare.

A summary of the performance of various thyroid cancer frameworks is detailed in Table 10.

**Table 10.** Evaluation of the performance of various thyroid cancer frameworks in percentages (%).

| Ref. | AI Model | Dataset | ACC | SPE | SEN | PPV | F1 | NPV | AUC |
|------|----------|---------|-----|-----|-----|-----|-----|-----|-----|
| [238] | CNN | PD | 88.00 | 79.10 | 98.10 | - | - | - | - |
| [103] | ELM | PD | 87.72 | 94.55 | 78.89 | - | - | - | - |
| [108] | MLP | PD | 87.16 | 87.05 | 91.18 | 16.20 | 27.50 | 99.70 | - |
| [142] | SVM | PD | 63.27 | 71.85 | 38.46 | 32.43 | - | 76.87 | - |
| [239] | RF | PD | 86.80 | 87.90 | 85.20 | - | - | - | 92.00 |
| [240] | LR | PD | 77.80 | 79.80 | 70.60 | - | - | - | 75.00 |
| [231] | B | PD | 84.69 | 86.96 | 82.69 | 87.76 | - | 81.63 | 88.52 |
| [241] | Ensemble DL | Cytological images | 99.71 | - | - | - | - | - | - |
| [100] | VGG-16 | Cytological images | 97.66 | - | - | - | - | - | - |
| [54] | VGG-16 | | 99.00 | 86.00 | 94.00 | - | 88.00 | - | - |
| [43] | RF | Ultrasound | - | - | - | - | - | - | 94.00 |
| [187] | k-SVM | Ultrasound | - | - | - | - | - | - | 95.00 |
| [131] | ANN | Ultrasound | - | - | - | - | - | - | 69.00 |
| [125] | SVM RF | Ultrasound | - | - | - | - | - | - | 95.10 |
| [242] | ANN SVM | Ultrasound | 96.00 | - | - | - | - | - | - |
| [243] | RF | Ultrasound | - | - | - | - | - | - | 75.00 |
| [244] | CNN | DICOM | 83.00 | 85.00 | 82.40 | - | - | - | - |
| [28] | CNN | DICOM | - | 91.50 | - | - | - | - | - |
| [117] | Fine-tuned DCNN | PD | 99.10 | - | - | - | - | - | - |

**Table 10.** *Cont.*

| Ref. | AI Model | Dataset | ACC | SPE | SEN | PPV | F1 | NPV | AUC |
|---|---|---|---|---|---|---|---|---|---|
| [245] | ResNet18-based network | PD | 93.80 | - | - | - | - | - | - |
| [246] | Multiple-scale CNN | PD | 82.20 | - | - | - | - | - | - |
| [99] | ThyNet | PD | - | - | - | - | - | - | 92.10 |
| [247] | Alexnet CNN | PD | 86.00 | - | - | - | - | - | - |
| [175] | DNN | ACR TIRADS | 87.20 | - | - | - | - | - | - |
| [124] | CNN (BETNET) | Ultrasound | 98.30 | - | - | - | - | - | - |
| [248] | ResNet | TIRADS | 75.00 | - | - | - | - | - | - |
| [249] | Xception | CT images | 89.00 | 92.00 | 86.00 | - | - | - | - |
| [120] | DCNN | Sonographic images | 89.00 | 86.00 | 93.00 | - | - | - | - |
| [250] | Google inception v3 | Histopathology images | 95.00 | - | - | - | - | - | - |
| [251] | Cascade MaskR-CNN | Ultrasound | 94.00 | 95.00 | 93.00 | - | - | - | - |
| [252] | VGG16 | Ultrasound | - | 92.00 | 70.00 | - | - | - | - |
| [253] | VGG19 | Ultrasound | 77.60 | 81.40 | 72.50 | - | - | - | - |
| [40] | VGG16 | Ultrasound | 74.00 | 80.00 | 63.00 | - | - | - | - |
| [189] | SVM CNN | Ultrasound | 92.50 | 83.10 | 96.40 | - | - | - | - |
| [254] | CNN | TIRADS | 85.10 | 86.10 | 81.80 | - | - | - | - |
| [255] | CNN | TIRADS | 82.10 | 85.00 | 78.00 | - | - | - | - |
| [256] | CNN | TIRADS | 80.30 | 80.10 | 80.60 | - | - | - | - |
| [257] | CNN | US | 83.00 | 47.00 | 65.00 | - | - | - | - |
| [258] | CNN | MRI | 79.00 | 80.00 | 65.00 | - | - | - | - |
| [259] | CNN | US | 97.00 | 84.10 | 89.50 | - | - | - | - |
| [260] | CNN | CT image | 84.00 | 73.00 | 93.00 | - | - | - | - |
| [261] | CNN | US | 77.00 | - | - | - | - | - | - |

The reported accuracy, sensitivity, and specificity of AI models in the literature may vary widely based on the dataset used, the quality of the data, and the methodology employed. AI models' effectiveness in a controlled experimental environment may not reflect their performance in a real-world clinical setting. Factors such as noise in the data, incomplete data, and changing clinical conditions can dramatically influence the outcome. Therefore, it is crucial to scrutinize the model's robustness and reliability under various conditions.

*Limitations and Open Challenges*

Despite the success of AI tools in thyroid cancer diagnosis, their limitations hinder the development of effective solutions, make their application costly, and limit their diffusion. To achieve precise thyroid cancer detection, it is crucial to centralize and securely store all relevant data in one location, unless you opt for federated learning (FL) techniques [262]. Then, algorithms must be developed to identify all forms of thyroid cancer. Every TCD includes a set of training images, test images, nodule plans, and classifications of nodule characteristics of diverse sizes [263]. The datasets must be regularly updated using MRI, CT scans, X-rays, and clinically obtained scans to assess thyroid conditions, and they should also include demographic information such as race, ethnicity, gender, and age. Additionally, it is important to establish a unified and centralized database accessible to all medical centers to test, validate, and apply AI algorithms to existing data [264]. Moreover, the rest of the limitations and open challenges can be summarized as follows:

- Insufficient clean data and accuracy: The lack of comprehensive and annotated datasets regarding the incidence and spread of cancer, specifically thyroid cancer, is a major hindrance to accurate cancer diagnoses and efficient treatment. Medical statistics often do not properly record the number of deaths caused by thyroid cancer, making data collection and validation challenging [265]. This results in a limited quantity of data typically collected from one center, due to the absence of a dedicated thyroid cancer clinical database shared among institutions. The accuracy of AI algorithms in diagnosing thyroid cancer is also limited by the scarcity of available labeled

cases for clinical outcomes [266]. Researchers acknowledge that a large quantity of data is necessary for a neural network to yield accurate results, but caution must be taken in regard to the data added during the learning phase, as it can introduce noise.

- Thyroid gland imaging: In the diagnostic evaluation of thyroid cancer, computed tomography (CT) and MRI are available options, but they are not considered the preferred methods due to their high cost and unavailability in certain cases [55]. Instead, ultrasound is commonly used as an alternative to physical exams, radioisotope scans, or fine-needle aspiration biopsies. During an ultrasound examination, the doctor is able to assess the activity of the gland by observing the echo of the node and determining its echogenicity, size, limits, and the presence of calcifications. However, the results obtained from ultrasound tests are not always accurate enough to differentiate between benign and malignant nodes and the images obtained can be more prone to noise [267].

- DL models' hyperparameters: Choosing the right DL algorithm is crucial in addressing various issues, particularly those related to thyroid cancer diagnosis. Due to the close similarities between benign and malignant tumors, as well as between tumors and other types of lymphocytes, it is challenging to differentiate between them accurately [268]. To achieve this, a significant increase in the number of layers for feature extraction may be required. However, this results in a longer processing time, especially when dealing with large quantities of data, which can impact the timeliness of the diagnosis for cancer patients [54].

- Computation cost and storage space: In the field of algorithms, time computing is a metric that assesses the computational complexity of an algorithm, which predicts the time it takes to run the algorithm by calculating the number of basic operations it performs, as well as its dependence on the size of the input. Typically, time computing is expressed as $O(n)$, where $n$ represents the size of the input, measured in terms of the number of bits required to represent it [269]. Researchers in the AI field, especially those working on thyroid cancer or other types of cancer diagnosis, face the challenge of finding algorithms that are both highly accurate and efficient in terms of processing time. They aim to develop algorithms that can analyze vast quantities of data quickly while still providing accurate results. Moreover, the volume of data used in these algorithms can sometimes exceed the available storage space [54].

- Imbalanced dataset: The distribution of cancer elements within categories related to thyroid tissue cells is often uneven, as these cells often make up a minority of the total tissue cell dataset. As a result, the dataset is highly imbalanced, consisting of both cancer cells and normal cells. This unbalanced distribution of features in cancer cell detection datasets often results in the suboptimal performance of AI algorithms used for the detection [270].

- Sparse labels: Labeling is a crucial aspect of computed tomography (CT) detection, specifically for distinguishing between normal and abnormal cancer cells. However, the process can be time-consuming and costly due to the limited number of available labels. This scarcity results in inconsistent decisions and can negatively impact the accuracy of AI algorithms, which heavily rely on labeled data. This can eventually undermine the trust and credibility of this type of application [270].

- The volume of data: At present, with the advancement in technology, especially in the field of thyroid cancer diagnosis and the growing volume of medical and patient data, researchers are facing challenges in suggesting algorithms that can effectively handle a limited number of samples, noisy samples, unannotated samples, sparse samples, incomplete samples, and high-dimensional samples. This requires AI algorithms that are highly efficient and capable of processing vast quantities of data exchanged between healthcare providers and patients or among specialist physicians [271].

- Error susceptibility: Despite AI being self-sufficient, it is still susceptible to errors. For instance, when training an algorithm with TCDs to diagnose cancerous regions, it can result in biased predictions if the training sets are biased. This can lead to a series of

incorrect results that may go unnoticed for an extended period. If detected, identifying and correcting the source of the problem can be a time-consuming process [272].

- Data form: Despite the numerous advancements in the use of AI for thyroid cancer detection, several limitations persist and pose a challenge to its progress. With the growing demand for various medical imaging technologies that result in vast quantities of data needed for AI algorithms, coordinating and organizing this information has become a daunting task. This can largely be attributed to the absence of proper labeling, annotation, or segmentation of the data, making it difficult to manage effectively [273].

- Unexplainable AI: The utilization of AI in the medical field can sometimes yield results that are unclear and lack proper justification, known as a "black box". This leaves doctors unsure about the accuracy of the results and may lead to erroneous decisions and treatments for patients with thyroid cancer. Essentially, AI can behave like a black box and fail to provide understandable explanations for its outputs [274].

- Lack of cancer detection platform: One of the major barriers to detecting various cancers, particularly thyroid cancer, is the limited availability of platforms for reproducing and examining previous results. This shortage represents a significant weakness and hinders the comparison of AI algorithm performance, making it challenging to improve their efficacy [159]. The presence of online platforms with comprehensive datasets, cutting-edge algorithms, and expert recommendations is vital in aiding doctors, researchers, developers, and specialists to make informed decisions with a low margin of error. Such platforms also provide a crucial supplement to clinical diagnoses by allowing for a more comprehensive experimentation and comparison [275].

- The digitization and loss data: The digitization of medical records has become a necessity, particularly in the realm of cancer diagnosis, due to the widespread adoption of various technologies such as whole-slide images. These latter serve as digital versions of glass slides, facilitating the application of AI techniques for pathological analysis [276]. Despite its benefits, digitization in the medical field is confronted with certain limitations, such as the risk of significant information loss during the quantification and inaccuracies that may arise from data compression utilized in autoencoder algorithms. Hence, it is crucial to be mindful in selecting the right digitization technology to preserve the information and maintain the originality of the data [277,278].

- Contrast: The absence of sufficient contrast in the tissues neighboring the TG complicates the process of accurately analyzing and diagnosing thyroid cancer.

## 8. Future Research Directions

We also highlight the future trajectory of AI in thyroid carcinoma detection, discussing emerging trends and technologies while considering their ethical implications. The ethical considerations do not end there, as we further examine issues related to data privacy, accountability, and equity. This section highlights promising research trends that will have a major effect on enhancing thyroid cancer detection in the future.

### 8.1. Explainable Artificial Intelligence (XAI)

The use of AI systems in decision-making is crucial, but they can be complex and difficult to understand. To address this issue, the field of XAI has emerged, which aims to provide transparency in AI models. The need for XAI is especially important in health applications where the interpretation of results is crucial. The use of XAI has been demonstrated in the analysis of incurable diseases affecting the TG, as seen in several studies such as [279–283]. The difference between AI and XAI is illustrated in Figure 7. In [171], the authors present an XAI model for the detection of thyroid cancer, which improves the confidence of medical practitioners in the predictions. Unlike traditional AI algorithms, XAI models provide evidence to support their conclusions and avoid the limitations of

"black box" algorithms. By using XAI, clinicians can make more informed decisions with greater confidence.

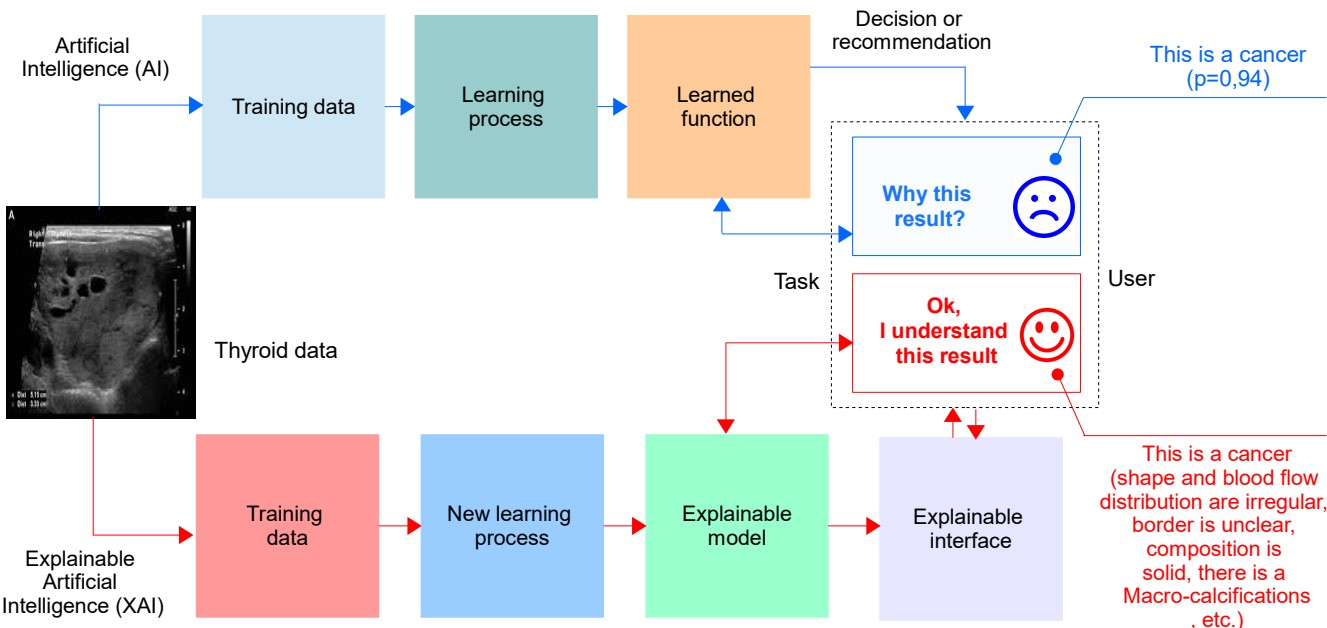

**Figure 7.** XAI diagram block.

## 8.2. Edge, Fog, and Cloud Computing for Implementation

The edge network is a combination of edge computing and AI that processes algorithms based on AI near the source of data [284]. This allows for better performance and lower costs for applications that require heavy information processing and reduces the need for long-distance communication between the patient and the doctor. The proximity of the information and storage capabilities to the end-user in the health sector allows for direct and immediate access [285]. To further enhance performance, the detection of thyroid cancer in edge networks relies on the use of fog computing, which is a decentralized computing architecture located between the cloud and the data-producing devices. This architecture allows for the flexible placement of computing and storage resources in logical locations, improving performance [286]. To ensure the proper operation of the AI-based thyroid cancer detection system, it utilizes cloud computing as an access point. This guarantees that the stored data, servers, databases, networks, and programs are accessible and shared among specialized doctors, as long as it is connected to the Internet. Such a hybrid system has proven to be effective for medical applications, including the detection of thyroid cancer, as seen in various studies including [287–297].

## 8.3. Reinforcement Learning (RL)

RL, a subfield of ML, allows agents to make decisions in interactive environments through trial and error, observation, and learning (as depicted in Figure 8). In recent years, there has been a significant interest in using RL for detecting incurable diseases and providing explanations to aid medical decision-making. For example, RL is used in [298] to classify cancer data, and deep RL is used in [299] to segment lymph node sets. The authors generate pseudo-ground-truths using RECIST-slices and achieve the simultaneous optimization of lymph-node bounding boxes through the interaction between a segmentation network and a policy network.

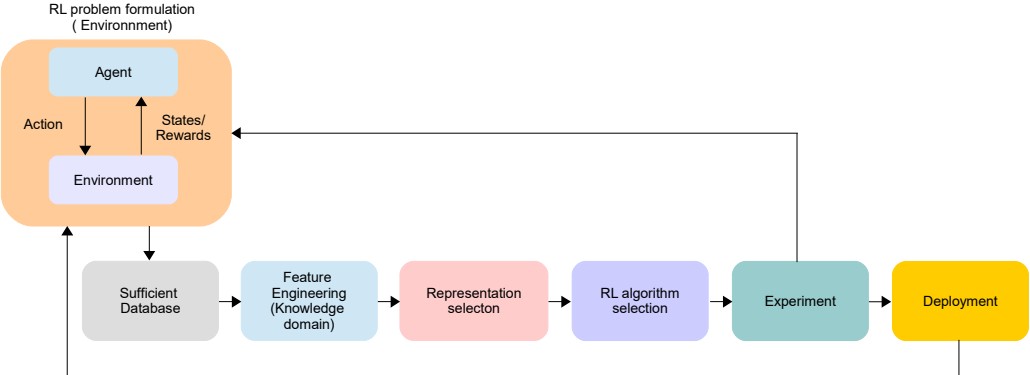

**Figure 8.** Deep RL procedure.

*8.4. Transfer Learning (TL)*

TL is a valuable solution to the overfitting and precision challenges faced by diagnosis systems [300–302]. This technique leverages stored knowledge from a specific problem to address other issues such as reducing training time and data volume [271,303]. Its use in the diagnosis of the TG is demonstrated in Figure 9. For instance, the Enhance-Net model, as introduced in [304], could serve as a source model for enhancing the performance of a target DL model designed for real-time medical images. Furthermore, in [158], the authors tackle the challenge of capturing appropriate features of benign and malignant nodules using CNNs. They transfer the knowledge learned from natural data to an ultrasound image dataset to produce hybrid semantic deep features. The TL technique has also been successfully applied to classify thyroid nodules images in [164]. Other related works can be found in [162,252,305–307].

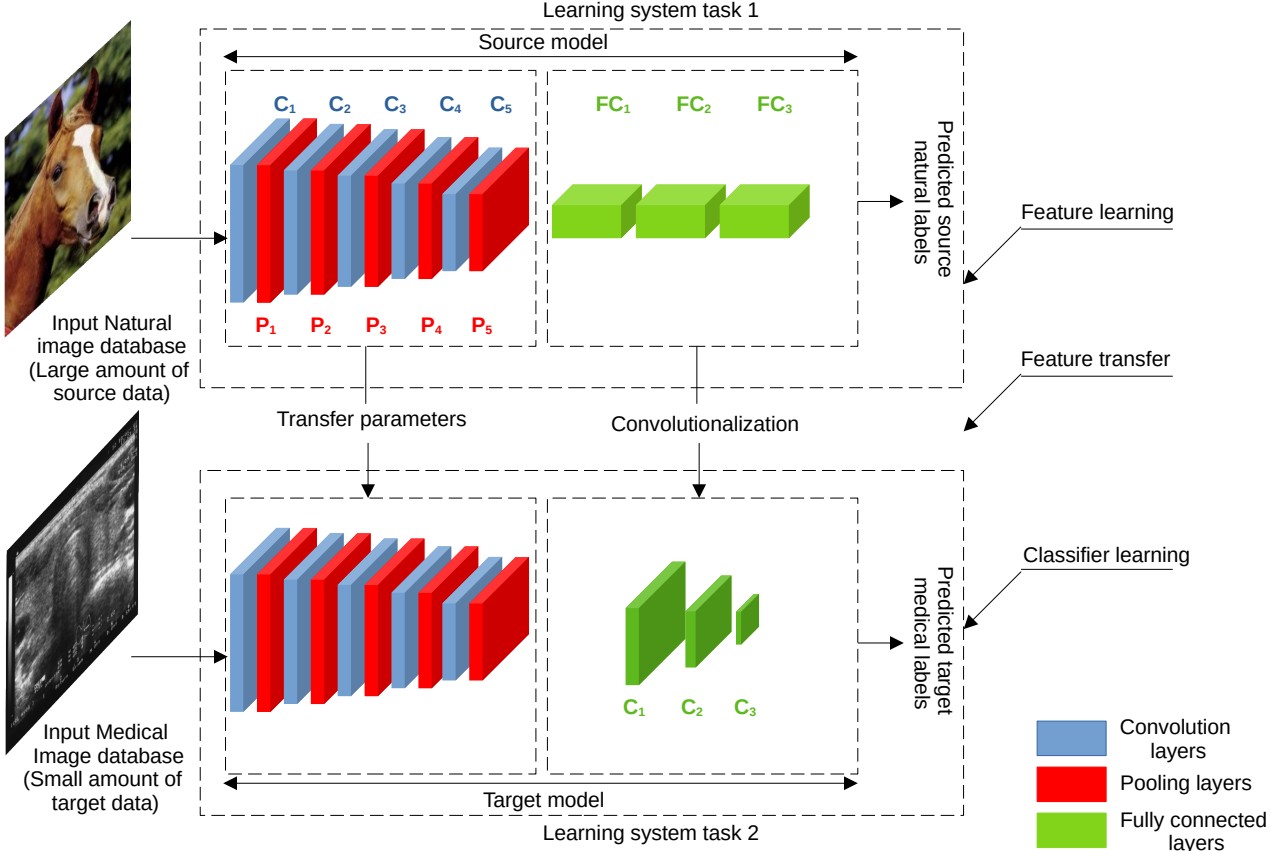

**Figure 9.** Deep transfer learning for thyroid diagnosis.

### 8.5. Panoptic Segmentation (PS)

The challenge of accurately separating and dividing objects with diverse and overlapping appearances remains an issue, particularly in the medical field. To address this, many researchers have put forth proposals for a comprehensive and cohesive segmentation of various details [308,309]. The focus has been on PS, which combines both instance and semantic segmentation to identify and separate objects. In semantic segmentation, the goal is to classify each pixel into specific classes, while in instance segmentation, the focus is on segmenting individual object instances. AI has been incorporated into this model through supervised or unsupervised instance segmentation learning, making it well suited for medical applications (Figure 10). This has been demonstrated in works such as [310,311].

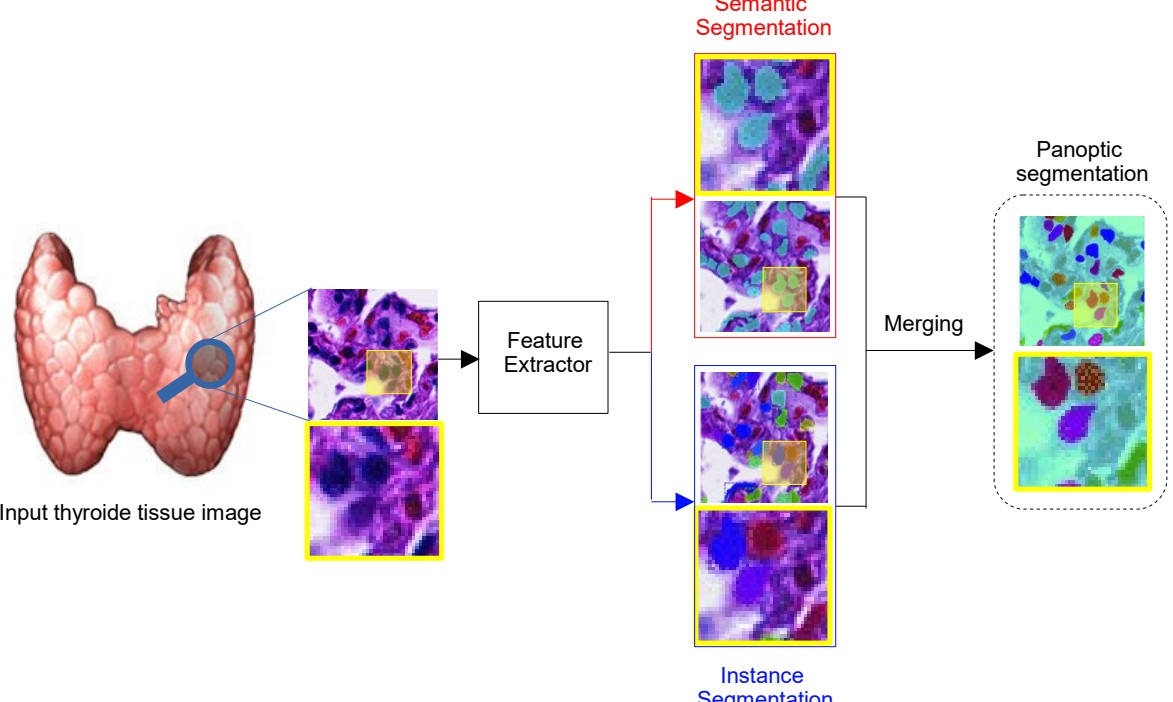

**Figure 10.** PS architecture.

### 8.6. Internet of Medical Imaging Things (IoMIT)

The IoMIT has recently gained widespread attention in the medical field, as it seeks to enhance healthcare delivery and reduce treatment costs through the exchange of health data between patients and doctors using connected devices with wireless communication (Figure 11). One example of this integration can be found in [312], which proposes an AI-based solution for the early detection of thyroid cancer in the IoMT, utilizing CNN to improve the differentiation between benign and malignant nodules, ultimately saving lives. Other relevant studies related to the IoMIT have also been conducted, such as [313,314].

### 8.7. Three-Dimensional Thyroid Cancer Detection (3D-TCD)

The conventional 2D ultrasound is widely used for diagnosing thyroid nodules, but its static images may not accurately reflect the nodule's structures. Hence, the use of 3D ultrasound has gained attention as it provides a more comprehensive view of the lesion by reconstructing its features and enabling a better differentiation between different diagnoses [315]. With the ability to examine complex growth patterns and margins and to give shape from multiple angles and levels, 3D ultrasound can provide a more accurate evaluation of the morphological features of thyroid nodules in comparison to 2D images. This has been confirmed through comparative studies between 3D and 2D ultrasound images [316–318].

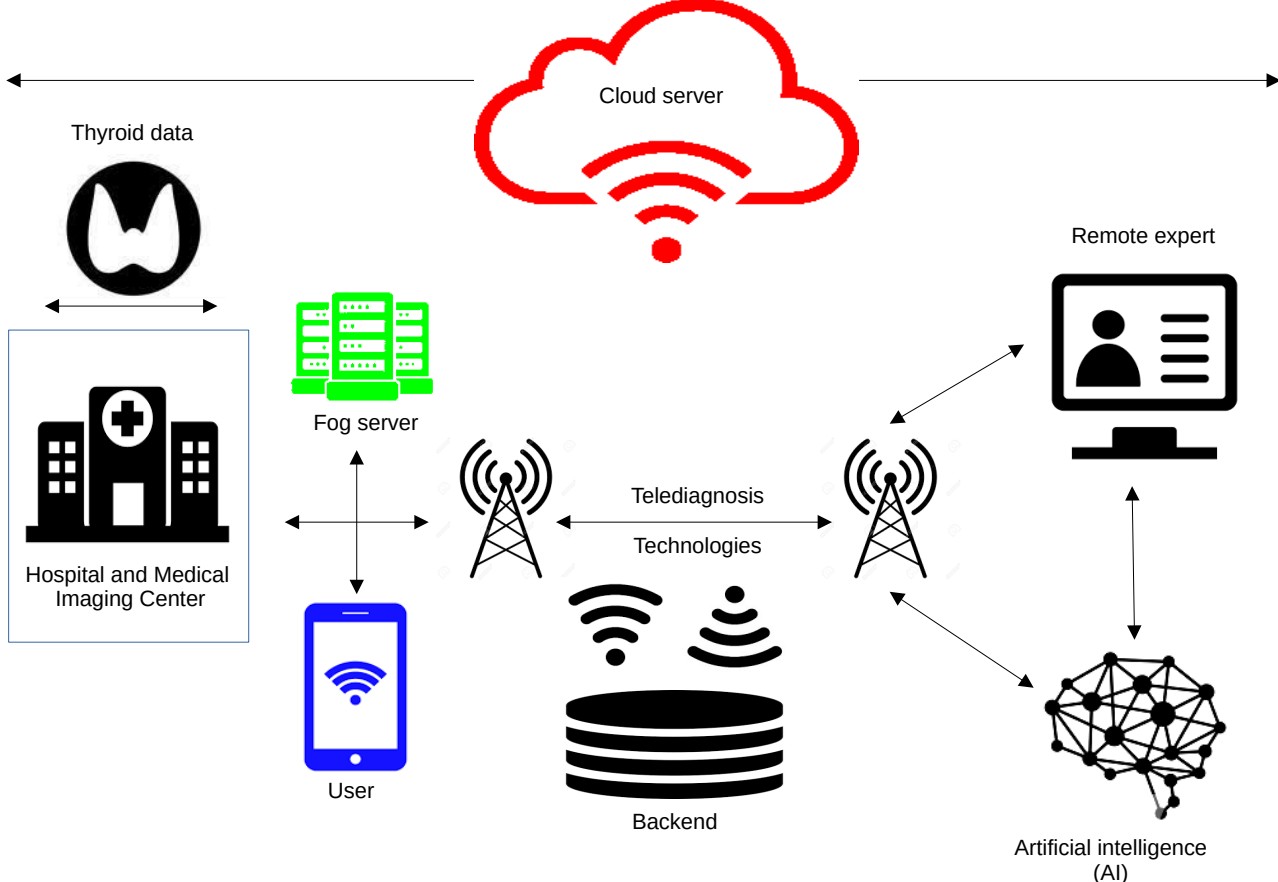

**Figure 11.** Example of a hybrid network system based on AI for thyroid cancer detection.

### 8.8. AI in Thyroid Surgery (AI-TS)

In light of the challenges faced in surgical procedures, the use of AI-powered robots in surgical practices is becoming increasingly essential. AI has the potential to address numerous clinical issues by analyzing and sharing massive quantities of data to support decisions with a level of accuracy comparable to that of healthcare professionals [319]. Companies are incorporating AI into surgical practices by training AI-based systems and providing robots that assist surgeons in operating rooms, supply surgical materials, handle contaminated materials and medical waste, remotely monitor patients, and collect and organize patient data such as electronic medical records, vital signs, laboratory results, and video footage [320]. As such, it is important for surgeons to have a strong understanding of AI in order to grasp its impact on healthcare. While AI-powered robotic surgery may still be some time away, collaboration across various fields can accelerate AI's capabilities and improve surgical care [321–328].

### 8.9. Wavelet-Based AI

Recently, the wavelet transform, specifically the first and second-generation ones, has gained recognition for its ability to detect various forms of cancer, especially when integrated with AI. This combination has become crucial in the medical field, providing doctors and surgeons with a tool to accurately diagnose diseases more efficiently and quickly [329,330]. The proposed method is based on preprocessing the dataset through DWT and then evaluating the performance of AI in classifying different types of tumors that can impact organs in the body (as explained in Figure 12). This model holds great potential for the detection of thyroid cancer and researchers are encouraged to test different wavelets available in the literature to further improve its effectiveness [331].

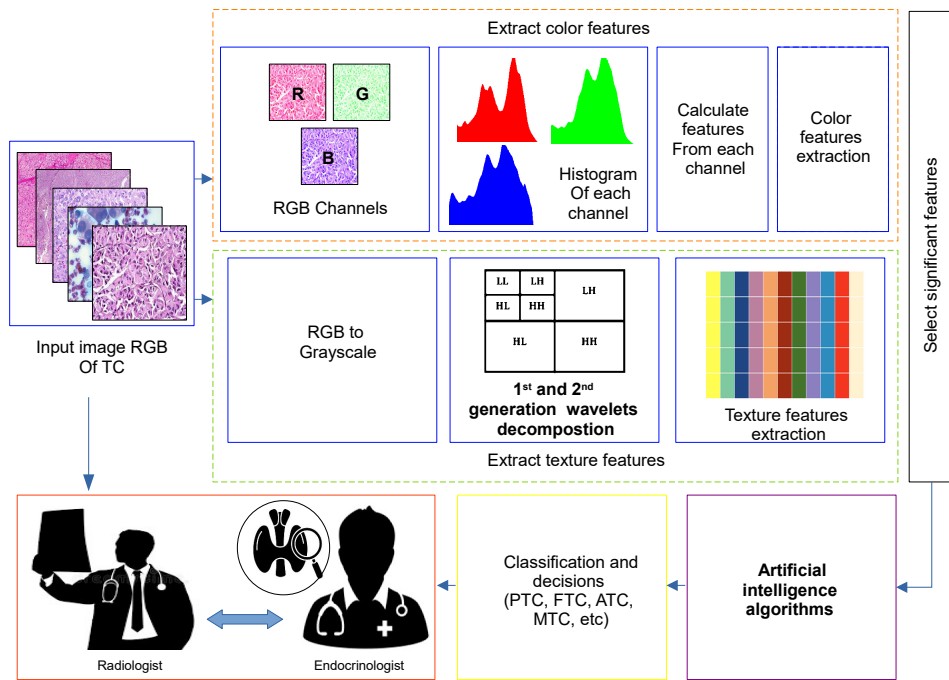

**Figure 12.** Applications of AI-based on wavelet in the detection of thyroid cancer.

### 8.10. Learning with Reduced Data

One of the hurdles in implementing AI in the medical sector is acquiring adequate data and annotations. AI's capability to minimize the need for labeled data in making an accurate diagnosis is crucial [332]. This can be achieved through various learning methods such as semi-supervised learning, supervised learning, USL, or alternative approaches that necessitate a smaller quantity of annotated data (Figure 13) [333].

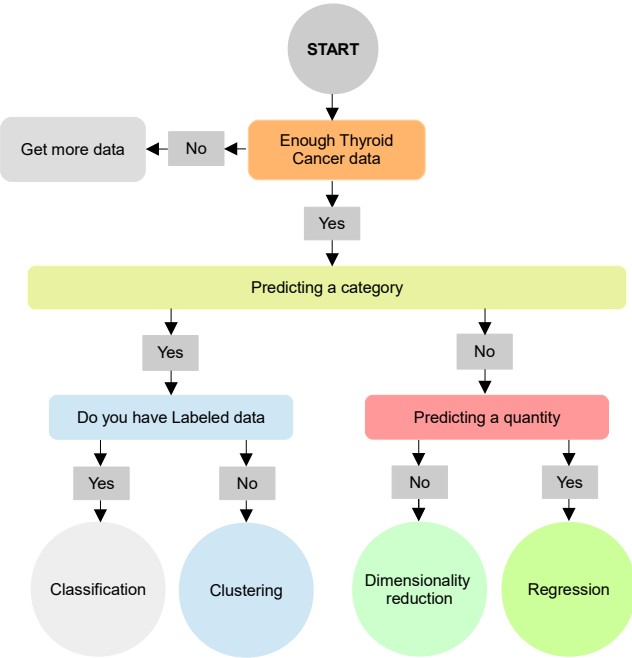

**Figure 13.** Diagram of the choice of AI-algorithms for thyroid cancer detection.

### 8.11. Recommender Systems (RSs)

The abundance of data collected from online medical platforms and electronic health records can make it challenging for thyroid cancer patients to access relevant and accurate

information [334]. The high cost of healthcare data also poses difficulties for doctors to track patients and manage a large patient volume with various treatment options. Given these challenges, the implementation of recommender systems (RSs) has been proposed to improve decision-making in healthcare and ease the workload for both patients and oncologists [335,336]. The use of RS in digital health provides personalized recommendations, an accurate analysis of big data, and stronger privacy protection through integration with AI and machine learning technologies [337] as depicted in Figure 14.

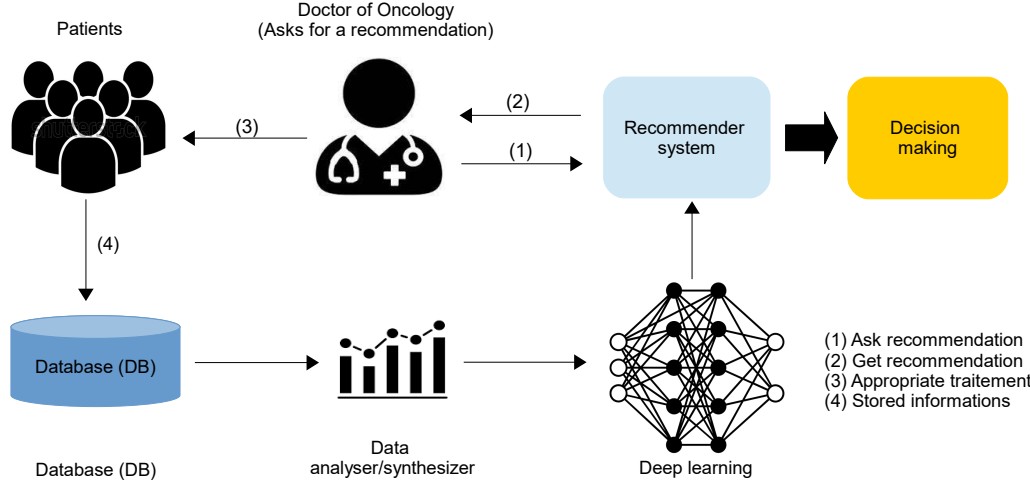

**Figure 14.** RSs for thyroid cancer detection.

*8.12. Federated Learning (FL):*

FL has become very popular in the field of healthcare applications [338]. The surrounding conditions greatly affect human health and cause negative effects on the economy. Diseases of the thyroid gland are among the most common health problems that have become noticeable among various groups of society in recent times. ML can play a vital role in such medical conditions, as the collected data can be exploited to train an ML model that can predict critical conditions. Emphasizing that patient data across different medical centers should be handled privately, the FL setup is the natural choice for such applications, as depicted in Figure 15. Therefore, in [339], the authors compared the performance of FL against five conventional deep learning methods (VGG19, ResNet50, ResNext50, SE-ResNet50, and SE-ResNext50) for analyzing and detecting TCDs.

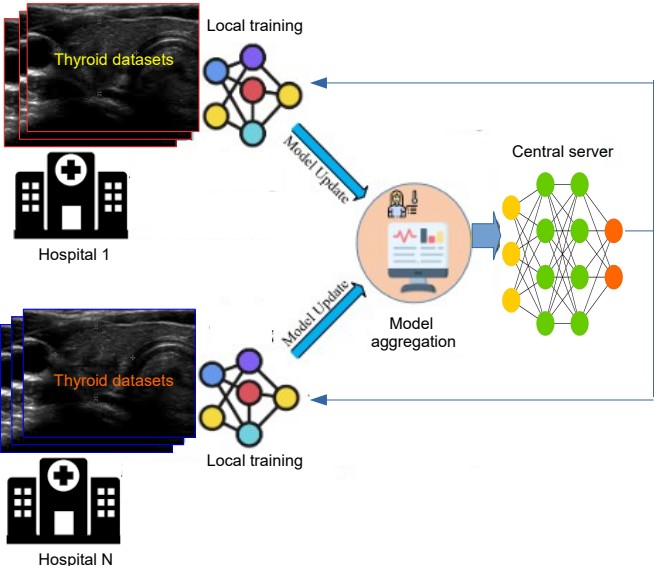

**Figure 15.** FL for healthcare.

*8.13. Generative Chatbots*

Most recently, the realm of AI has witnessed significant advancements, particularly in the development of generative chatbots and large language models such as GPT variants [340]. These state-of-the-art models, trained on vast quantities of data, are adept at generating humanlike text and engaging in coherent conversations, going beyond mere predefined responses. As their capability has expanded, so too has their potential for application across various domains, healthcare being one of the prominent ones. In the healthcare sector, these sophisticated models are being explored for patient engagement, preliminary symptom checks, providing health-related information, and even assisting professionals with medical research and data analysis [3]. The integration of such technology holds the promise of streamlining healthcare processes, enhancing patient experience, and augmenting the capabilities of healthcare professionals, albeit with the necessary precautions and ethical considerations in place [341].

Using generative chatbots or models such as ChatGPT to diagnose thyroid cancer (or any medical condition) directly would be inappropriate and potentially dangerous. However, they can be incorporated into healthcare settings in auxiliary roles [342]. Typically, chatbots can gather preliminary information from patients, including their symptoms, family history, and lifestyle habits. These data can provide a better understanding of the patient's concerns before they meet a healthcare professional. Moreover, they can be programmed to provide information about thyroid cancer, such as risk factors, symptoms, and preventive measures [343]. Patients can learn about the disease and its potential signs, allowing them to approach healthcare providers if they find any matching symptoms. Furthermore, while they cannot replace professional diagnostic tools, they can be designed to guide users through a series of questions that could highlight potential risk factors or symptoms, encouraging them to consult a medical professional for a more comprehensive evaluation [344].

On the other hand, once a diagnosis has been made, chatbots can provide patients with information on treatment options, side effects, diet recommendations, and answer frequently asked questions. Additionally, they can (i) remind patients to take their medications, attend follow-up appointments, or perform regular self-examinations or monitoring, (ii) offer support in terms of relaxation techniques, provide resources for further psychological support, or even just offer a nonjudgmental "listening ear", and (iii) assist doctors and other healthcare professionals by providing instant information about thyroid cancer, recent research, or treatment options, acting as a dynamic reference tool [345].

## 9. Conclusions

In this research, a comprehensive overview of DNNs was presented, spotlighting their ascendant trend in recent years owing to their superior accuracy compared to other methodologies. An array of algorithms and training structures, inclusive of their benefits and constraints, was delineated. DNNs are manifestly pivotal in myriad real-world applications, particularly lauded for their generalizability and tolerance to noise.

Notwithstanding these advancements, barriers persist in fully adopting DNNs in thyroid cancer detection. A paramount obstacle is the absence of clean datasets and platforms. To cultivate efficient and formidable cancer detection models capable of discerning more advanced malignancies, these data constraints warrant meticulous attention.

Future research needs to be oriented towards circumventing these impediments and enhancing thyroid cancer detection's caliber. Furthermore, this study underscores the urgency for amplified research endeavors in thyroid cancer identification, especially given the diagnostic precision coveted by medical practitioners. While the detection of various cancers in two or three dimensions is a burgeoning research area, the deficiency in expertise with diverse geometric transformations and the requisite dimensional databases curtails the precision in diagnosing terminal illnesses. Hence, pioneering methodologies to discern disparate magnitudes of cancerous nodules become indispensable. Such innova-

tions can exponentially augment treatment velocity, diagnostic accuracy, enable proactive epidemiological surveillance, and subsequently mitigate mortality rates.

Emerging technologies, namely explainable AI, edge computing, RL, PS, and RSs, have unfurled novel research horizons in thyroid cancer detection. These innovations are proving invaluable for clinicians by streamlining the diagnostic process, curtailing detection time frames, and fortifying patient confidentiality. As a trajectory for future endeavors, our focus will pivot towards a deeper probe into the contributions of these avant-garde technologies. Our ambition is to foster a seismic paradigm shift in cancer detection by ideating state-of-the-art, privacy-centric technologies for thyroid cancer patient identification and broader applications, such as telehealth.

**Author Contributions:** Conceptualization, Y.H. (Yassine Habchi); methodology, Y.H. (Yassine Habchi), Y.H. (Yassine Himeur) and A.B.; software, Y.H. (Yassine Habchi); validation, Y.H. (Yassine Habchi), Y.H. (Yassine Himeur) and A.B.; formal analysis, Y.H. (Yassine Habchi), Y.H. (Yassine Himeur) and H.K.; investigation, A.O.; data curation, Y.H. (Yassine Habchi); writing—original draft preparation, Y.H. (Yassine Habchi); writing—review and editing, Y.H. (Yassine Himeur), H.K., A.B., S.A., A.C., A.O. and W.M.; visualization, Y.H. (Yassine Habchi) and A.C.; supervision, Y.H. (Yassine Himeur) A.B. and W.M. All authors have read and agreed to the published version of the manuscript.

**Funding:** This work was supported by the Laboratory of Energetic System Modelling (LMSE) of the University of Biskra, Algeria, under the patronage of the General Directorate of Scientific Research and Technological Development (DGRSDT) in Algeria. The research project was approved by the Ministry of Higher Education and Scientific Research in Algeria, under number A01L08UN070120220003. Open-access funding was provided by the University of Dubai.

**Institutional Review Board Statement:** Not applicable.

**Data Availability Statement:** Not applicable.

**Conflicts of Interest:** The authors declare no conflict of interest.

## Abbreviations

| | |
|---|---|
| AC | Active contour |
| AI | Artificial intelligence |
| ANN | Artificial neural network |
| ATC | Anaplastic thyroid carcinoma |
| BA | Bootstrap aggregation |
| Bi-LSTM | Bidirectional LSTM |
| BN | Bayesian network |
| CAD | Computer-aided diagnosis |
| CFS | Correlation-based feature selection |
| CNN | Convolutional neural network |
| CT | Computed tomography |
| DAE | Denoising autoencoder |
| DCG | Discounted cumulative gain |
| DCNN | Deep convolutional neural network |
| DDTI | Digital Database Thyroid Image |
| DL | Deep learning |
| DNN | Deep neural network |
| DR | Dimensionality reduction |
| DT | Decision trees |
| DTCW | Double-tree complex wavelet transform |
| DWT | Discrete wavelet transfer |
| ELM | Extreme learning machine |
| ER | Error rate |
| FB | Feature bagging |
| FL | Federated learning |
| FNAB | Fine-needle aspiration biopsy |

| | |
|---|---|
| FTC | Follicular thyroid carcinoma |
| GAN | Generative adversarial network |
| GEO | Gene expression omnibus |
| GLCM | Gray-level co-occurrence matrix |
| HOG | Histogram of oriented gradient |
| ICA | Independent component analysis |
| IG | Information gain |
| IoMIT | Internet of medical imaging things |
| KM | K-means |
| KNN | K-nearest neighbors |
| LBP | Local binary patterns |
| LR | Logistic regression |
| LSTM | lLong short-term memory |
| ML | Machine learning |
| MLP | Multilayer perceptron |
| MRI | Magnetic resonance imaging |
| MRM | MicroRNA regulatory module |
| MRR | Mean reciprocal rank |
| MSE | Mean squared error |
| MTC | Medullary thyroid carcinoma |
| NCDR | National Cancer Data Repository |
| PCA | Principal component analysis |
| PLCO | Prostate, Lung, Colorectal, and Ovarian |
| PM | Probabilistic models |
| PS | Panoptic segmentation |
| PSNR | Peak signal to noise ratio |
| PTC | Papillary carcinoma |
| RBF | Radial basis function |
| RBM | Restricted Boltzmann machine |
| RF | Random forest |
| RL | Reinforcement learning |
| RMSE | Root-mean-square error |
| RNN | Recurrent neural network |
| RS | Recommender systems |
| SEER | Surveillance, Epidemiology, and End Results |
| SL | Supervised learning |
| SVM | Support vector machine |
| TCD | Thyroid cancer dataset |
| TCGA | The Cancer Genome Atlas |
| TCL | Traditional classification |
| TD | Thyroid disease |
| TDDS | Thyroid Disease Data Set |
| TG | Thyroid gland |
| TIRADS | Thyroid Imaging, Reporting, and Data System |
| TI-RADS | Thyroid Imaging, Reporting, and Data System |
| TL | Transfer learning |
| TN | Thyroid nodules |
| USL | Unsupervised learning |
| XAI | Explainable AI |
| XAI | Explainable artificial intelligence |
| XGBoost | Gradient tree boosting |

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
