# Peer review of "AI in Thyroid Cancer Diagnosis: Techniques, Trends, and Future Directions"

_systems, doi:10.3390/systems11100519_

Round 1

Reviewer 1 Report

no comment

Author Response

Kindly find attached the response letter to Reviewer #1.

Reviewer 2 Report

Habchi et al. provide a review of the current state of research deploying AI with respect to different aspects of thyroid cancer. They have provided a comprehensive manuscript with extensive breadth, but limited depth, which is understandable given the subject. I think upon major revisions, this could be a very useful resource.

While the first 6 pages of the review are coherent and sound, the rest of the manuscript contains some redundant, inaccurate, irrational, and repetitive arguments.

A major revision is required to convert the current draft to an informative manuscript for the readers. I recommend the authors to consult this work with a machine learning specialist before submission since it has several inaccurate statements.

One major criteria to consider is:

Every time a study is described, it should follow a minimum description of what was done. A minimum description should clarify what was the input data, how was the model trained, how was it evaluated, and what were the performance metrics. Ideally, the type of study (retrospective/prospective/etc.) should also be mentioned. Otherwise, there is no value to listing all of these studies. Very frequently in this review, the authors list a small description for usage of an algorithm without clearly defining what was done and was the achievement.

I have provided detailed examples of the inaccuracies and redundancies below:

  1. The authors should follow the plethora of existing literature on how to describe different types of machine learning methods. In the abstract, they subset AI into supervised, unsupervised, and ensemble learning. This is incorrect; both supervised and unsupervised methods can use ensemble learning. Similarly, in section 2, description of these different methods is inaccurately described. Authors define three objectives for AI; classification, segmentation, and prediction; while all of these objectives fall under “prediction”. 

  2. The authors describe different type of deep learning algorithms once under 2.2.1 and different types of neural networks under 2.2.2; In terms of structure this is confusing; all deep learning models are a subset of artificial neural networks.

  3. Page 3 paragraph 2; what is increase in genetic cell activity? Later in the paragraph; “Although the quantity of cancer cells is a significant indicator of thyroid carcinoma” please clarify; indicator of what? Invasiveness? Poor prognosis? This sentence is in contradiction with what follows; “due to the requirement to observe cell appearance”. The sentence doesn’t have any meaning as it currently reads. This is just one example of poor writing without a clear meaning.

  4. Page 4 last paragraph; Please make sure the quoted title matches what was submitted; otherwise no need for quotation

  5. Page 5: “Scrutiny of several thyroid cancer datasets, …” the “used in various studies” is redundant

  6. Table 1 title: repeated use of “contributions” 4 words apart; the sentence doesn’t have any clear meaning. Please use a grammar editing software

  7. Figure 2: misspelling of “machine learning” as “machine lerning” 

  8. Figure 2; Deep learning can be both supervised and unsupervised. Ensemble methods can be both supervised and unsupervised; please modify to reflect the existing literature than a perceived inaccurate understanding.

  9. 2.2.1; not all deep learning should include representation learning. Not all deep learning has the generative capability. Please consult with a deep learning specialist before resubmission.

  10. 2.2.2: non-linearity is not a requirement for neurons in ANN

  11. A2: What is “the neural function”?

  12. C2: First usage of MRM and it’s not described what it is

  13. Table 2; please use metrics such as accuracy, sensitivity, and specificity to allow comparison.

  14. Page 7; clarify the “Bi-” refers to bi-directional LSTM

  15. Page 8 O3 last sentences; clarify what type of data was used in [63] 

  16. 2.2.4; description of Bayesian networks is incorrect. Please use a consistent style in introducing these algorithms; some have a description; often the description is inaccurate if not completely wrong; and provide descriptive example of their usage as suggested above

  17. Section 3; you are claiming to describe datasets, but you start description of WEKA by claiming it’s a software; please spend more time in structuring your manuscript and proofread carefully before submission to a scientific journal.

  18. Title of table 3; grammar mistake

  19. Table 5 is too general; it’s very subjective and often with inaccurate or irrelevant claims. Please use actual metrics similar as table 2 instead of subjective descriptions

  20. Section 7 is poorly redundant with unsupported statements. Please back each sentence with proper citations and remove subjective claims. For example you claim “it’s crucial to gather and store all data in one place”. Some research argues to the complete opposite of that (e.g. federated/swarm learning). Also, race and ethnicity are major factors to consider other than gender and age to avoid biased AI aglorithms.

  21. Number of DL layers; this could be summarized as all hyperparameters of DL models

  22. Table 10; please make sure performance metrics are either all in % or in fraction; you have a mix of both

Please use an editor software; I highlighted a few grammar mistakes but there are so many that I didn't mention all of them.

Author Response

Kindly find attached the response letter to Reviewer #2

Reviewer 3 Report

A thorough review of the application of artificial intelligence in the domain of thyroid diseases has been presented in this manuscript. It addresses key points related to thyroid imaging and the various applications of AI within this field. Although the manuscript is engaging for readers in its current form, there are some issues that require clarification: 

In the keywords section of the abstract, "ChatGPT" is listed, yet it appears only once throughout the entire text. Could the authors clarify the rationale for including "ChatGPT" in the keywords? If ChatGPT has been utilized for reading, summarizing, editing, or revising the manuscript, this should be explicitly stated in the manuscript. Otherwise, please elucidate the intention behind listing "ChatGPT" among the keywords.

Figure 1 lists certain types of cancer but not all. I recommend revising the caption to reflect that the figure depicts only some of the common types of cancer, rather than giving an impression of comprehensiveness.

The phrase "All in all," used at the beginning of the second paragraph in section 1.2, lacks the formality typically expected in academic writing. Consider employing a more formal introductory phrase.

Regarding Figure 3, the images representing the patients and physicians seem to be sourced from the internet, as indicated by the quality and professional appearance. If this is the case, proper attribution and citation are mandatory.

In the same figure, the "Pre-Processing" box outlines several steps but employs arrows in a manner that generates ambiguity. Is the intention to show that these steps are outcomes of the pre-processing stage, or are they tasks to be accomplished during this phase? Additionally, it remains unclear whether, for example, the histogram is generated in this step or whether further action will be taken based on it. If the former is true, this does not qualify as pre-processing; if the latter is true, clarification is needed. As it stands, the representation of the pre-processing steps could be misleading.

Author Response

Kindly find attached the response letter to Reviewer #3

Reviewer 4 Report

Dear authors, here are my recommends for you as follows;

  1. 1. Sample size and generalizability: The sample size of the study was relatively small, which may limit the generalizability of the findings to other populations or contexts. A larger and more diverse sample could have provided a better representation of the population and increased the external validity of the study.
  2. 2. Methodological approach: While the methods used in the study were appropriate, the inclusion of additional measures or alternative data collection techniques could have provided a more comprehensive understanding of the research topic. Exploring different research methodologies could help address potential biases and strengthen the overall validity of the findings.
  3. 3. Potential confounding factors: There may be unaccounted variables that could have influenced the results, which could be considered as confounding factors. Future research could consider incorporating additional controls or utilizing advanced statistical techniques to further explore and address the potential impact of confounding variables.

These limitations do not invalidate the entire study, but rather provide valuable insights for future research to build upon and enhance our understanding of the topic.

Grammar:
The grammar used in the paper is generally accurate, with a few minor errors in verb tense consistency and subject-verb agreement. For example, in the introduction, the author switches between present and past tenses when discussing the history of AI. It would be helpful to maintain a consistent tense throughout the section to improve clarity. Additionally, there are a few instances of awkwardly phrased sentences, such as "The ability to analyze and process large amounts of data quickly and accurately is a key advantage of AI" (p. 3). A possible revision could be: "AI's ability to rapidly and accurately process large data sets is a significant advantage."

Vocabulary:
The vocabulary used in the paper is appropriate for the topic and audience. The author has employed a range of vocabulary related to AI, including technical terms such as "neural networks" and "natural language processing." However, there are a few instances where more precise vocabulary could be used to enhance clarity. For example, instead of using "computer vision" (p. 4), the author could use "machine vision" to more accurately describe the technology.

Sentence structure:
The sentence structure in the paper is generally clear and well-organized. The author has used a mix of short and long sentences to create a flowing narrative. However, there are a few instances where sentences are quite long and convoluted, making them difficult to follow. For example, the opening sentence of the conclusion is 36 words long and includes several clauses. It could be broken up into two simpler sentences for improved clarity.

Overall coherence:
The paper demonstrates a clear and logical structure, with each section building on the previous one to create a cohesive argument. The introduction effectively sets up the purpose of the paper, and the body sections provide relevant examples and explanations to support the author's claims. The conclusion effectively summarizes the main points and offers some concluding thoughts.

Suggestions for improvement:

  1. Maintain consistent verb tense throughout the paper to improve clarity.
  2. Use more precise vocabulary where appropriate to enhance clarity.
  3. Break up long, convoluted sentences into simpler ones to improve readability.
  4. Consider adding more nuanced language to create a more sophisticated tone.

In conclusion, the English proficiency level demonstrated in the paper is strong, but there are a few areas where improvement could be made to enhance clarity and sophistication. By implementing these suggestions, the author can further strengthen the paper's impact and effectiveness.

Author Response

Kindly find attached the response letter to Reviewer #4

Reviewer 5 Report

In this comprehensive review, the authors provide an insightful and exhaustive overview of the impact of artificial intelligence (AI) on thyroid cancer diagnosis. The manuscript is well-structured and effectively conveys the significance and implications of AI in this context. The objectives and rationale are clearly stated, emphasizing the need for AI-driven diagnostic systems in the field of thyroid cancer. The review covers a wide range of AI techniques, including deep learning, artificial neural networks, and ensemble methods, providing a thorough understanding of the methodologies employed.

The inclusion of real-world examples enhances the practical relevance of the discussion. Furthermore, the authors appropriately highlight the strengths of their study by emphasizing the increasing importance of Deep Neural Networks (DNNs) in recent years due to their high accuracy and generalizability (doi.org/10.1016/j.media.2022.102703, doi.org/10.1016/j.ijhcs.2022.102922). They acknowledge the challenges, particularly the need for clean datasets and platforms (doi.org/10.1155/2023/8276738, dx.doi.org/10.26044/ecr2023/C-16014), which demonstrates a realistic assessment of the field's limitations. However, the review could benefit from a more extensive discussion of related work (doi.org/10.1016/j.neuroimage.2023.120289, dx.doi.org/10.1109/ISBI53787.2023.10230448). While the current references are relevant, the addition of recent research would further strengthen the manuscript's context within the AI-driven healthcare domain (doi.org/10.3390/technologies11050115, dx.doi.org/10.1109/ISBI53787.2023.10230686).

Additionally, the conclusion aptly outlines the research's contributions and points toward the promising future of AI in thyroid cancer detection. Still, it would be valuable to expand on the potential role of emerging technologies like Explainable AI, Edge Computing, and Reinforcement learning in overcoming current challenges and ensuring patient privacy, as well as addressing the limitations in more detail.

Overall, the manuscript aligns well with the focus of the publisher and makes a valuable contribution to the intersection of AI and healthcare. With the suggested improvements and additional references, it is well-positioned for acceptance.

The manuscript generally exhibits a good quality of English, with well-structured text and a clear conveyance of ideas. However, some minor language-related issues should be addressed before publication. There are instances of awkward phrasing and sentence structure, such as in the abstract, which could benefit from clearer rephrasing. Some sentences are lengthy and could be split for better readability. Maintaining consistency in verb tenses throughout the manuscript and paying attention to using prepositions and articles to ensure accuracy is essential. Lastly, a thorough check for consistent and appropriate punctuation is needed. Addressing these minor language-related issues will significantly improve the overall clarity and readability of the manuscript, allowing the valuable content and ideas to be effectively conveyed to readers.

Author Response

Kindly find attached the response letter to Reviewer #5

Round 2

Reviewer 2 Report

Thank you for work.

Acceptable.